# Benchmarking microbarom radiation and propagation model against infrasound recordings: a vespagram-based approach

Ekaterina Vorobeva[1,2], Marine De Carlo[3,4], Alexis Le Pichon[3], Patrick Joseph Espy[1], and Sven Peter Näsholm[2,5]

[1]Department of Physics, Norwegian University of Science and Technology, Trondheim, Norway
[2]NORSAR, Kjeller, Norway
[3]CEA, DAM, DIF, F-91297, Arpajon, France
[4]Univ. Brest, CNRS, IRD, Ifremer, Laboratoire d'Océanographie Physique et Spatiale (LOPS), IUEM, Brest, France
[5]Department of Informatics, University of Oslo, Oslo, Norway

**Correspondence:** Ekaterina Vorobeva (ekaterina.vorobeva@ntnu.no)

**Abstract.** This study investigates the use of a vespagram-based approach as a tool for multi-directional comparison between simulated microbarom soundscapes and infrasound data recorded at ground-based array stations. Data recorded at the IS37 station in northern Norway during $2014 - 2019$ have been processed to generate vespagrams (velocity spectral analysis) for five frequency bands between 0.1 and 0.6 Hz. The back-azimuth resolution between vespagrams and the microbarom model is harmonized by smoothing the modelled soundscapes along the back-azimuth axis with a kernel corresponding to the frequency-dependent array resolution. An estimate of similarity between the output of the microbarom radiation and propagation model and infrasound observations is then generated based on the image processing approach of the mean-square difference. The analysis reveals that vespagrams can monitor seasonal variations in the microbarom azimuthal distribution, amplitude, and frequency, as well as changes during sudden stratospheric warming events. The vespagram-based approach is computationally inexpensive, can uncover microbarom source variability, and has potential for near-real-time stratospheric diagnostics and atmospheric model assessment. Keywords: infrasound, vespa, microbaroms, array signal processing, stratosphere, atmospheric models

## 1 Introduction

Microbaroms are infrasound waves with frequencies typically between 0.1 and 0.6 Hz generated by non-linear interaction between counter-propagating ocean waves. Once generated, microbaroms penetrate the atmosphere where vertical wind and temperature gradients are responsible for the presence of waveguides or sound channels (Brekhovskikh, 1960; Diamond, 1963). Waveguides duct the infrasound between the ground and different atmospheric layers and are usually classified into tropospheric, stratospheric and thermospheric (Hedlin et al., 2012; de Groot-Hedlin et al., 2010). Seasonal variations in the zonal stratospheric wind (eastward - westward) are crucial for the stratospheric waveguide, which is of particular interest in this study. These

variations combined with an increase in temperature in the stratosphere make the effective sound speed to be higher than at the surface. This causes the refraction of the infrasound waves back to the ground. The low-frequency microbaroms can be ducted over long distances due to the weak attenuation, which is proportional to the frequency squared. Hence there is a potential to exploit microbaroms to probe the dynamics of the stratosphere, where the representation of the atmospheric dynamics in model products is often poorly constrained (Polavarapu et al., 2005; Rienecker et al., 2011; Smith, 2012; Amezcua et al., 2020).

The term "microbarom" was established by Benioff and Gutenberg (1939) who described quasi-continuous pressure fluctuations with periods of $0.5 - 5$ s recorded by two electromagnetic barographs installed by the Seismological Laboratory, California Institute of Technology, Pasadena, USA. Following Benioff and Gutenberg (1939), several microbarom studies were performed by scientists around the globe. Joint observation of microbaroms and microseisms (quasi-continuous fluctuations of the ground displacement generated by ocean waves) in California, USA (Gutenberg and Benioff, 1941), Christchurch, New Zealand (Baird and Banwell, 1940), Fribourg, Switzerland (Saxer, 1945, 1954; Dessauer et al., 1951) and New York, USA (Donn and Posmentier, 1967) demonstrated that the microbarom signals originate from the ocean.

Thereafter, efforts were made to develop theories to explain the physical mechanisms of microbarom generation (Brekhovskikh et al., 1973; Waxler et al., 2007). A recent model proposed by De Carlo et al. (2020) unifies aforementioned theories of microbarom generation, taking into consideration both the finite ocean-depth and the source radiation dependence on elevation and azimuth angles. This model can predict location and intensity of the source when coupled with an ocean wave spectrum model. However, for comparison with infrasonic observations at distant ground-based stations, it is necessary to consider the influence of the atmospheric structure on the microbarom propagation and ducting. This can, for example, be estimated using a semi-empirical range-dependent attenuation model in a horizontally homogeneous atmosphere (Le Pichon et al., 2012), or wave propagation simulation using 3-D ray tracing (Smets and Evers, 2014). Details on our suggested vespagram-based comparison approach to microbaroms modeled by a state-of-the-art microbarom radiation theory (De Carlo et al., 2020) are presented in Sect. 2.2.

In array signal processing, velocity spectral analysis (vespa) is an approach which analyzes recorded signals in terms of signal power as a function of time (Davies et al., 1971; Rost and Thomas, 2002; Schweitzer et al., 2012). The power is evaluated either at a fixed slowness, i.e. a constant apparent velocity with varying back-azimuth — corresponding to a circle in the slowness space — or at a fixed back-azimuth with varying apparent velocity — corresponding to a line in slowness space. The vespa power estimate can therefore be visualized as an image, called vespagram, with time on one axis and either back-azimuth (for a fixed apparent velocity) or apparent velocity (for a fixed back-azimuth) as the other axis.

Lonzaga (2015) used a phase diagram approach to demonstrate that infrasound arrivals from stratospheric ducts typically have apparent velocities between 340 and 380 m/s. In the current work, the main focus is on microbaroms. These low-frequency waves have an apparent velocity of around 350 m/s as found by Rind et al. (1973). As such, time-back-azimuth vespagrams estimated from infrasound array data at an apparent velocity of 350 m/s are used in the current study. Histograms of the apparent velocity statistics for the current dataset are provided in Appendix A, which also support the use of this apparent velocity when generating the vespagrams. For a given frequency band, such vespagrams can be compared in a straightforward manner to microbarom soundscapes modeled for a station location, after applying a smoothing kernel which harmonizes the

resolution given by the array response function main lobe and the microbarom model output. Both the vespagram and the microbarom model provide power estimates as a function of time and back-azimuth. These can be displayed as an image, and we utilize an image comparison approach based on the mean-square error to benchmark the model against vespagrams. In this study, 6 consecutive years of infrasound observations between 2014 and 2019 at a ground-based infrasound array located in Bardufoss, Norway (69.07° N, 18.61° E), denoted IS37 or I37NO (Fyen et al., 2014), are considered. An overview of the station configuration, data, and analysis methods is provided in Sect. 2.1.

The proposed vespagram-based approach is computationally low-cost and can monitor microbarom source variability over a year (Sect. 3.1) as well as detect changes during extreme atmospheric events such as sudden stratospheric warmings (Sect. 3.2). It might be further refined for applications such as near-real-time diagnostics of ocean wave and atmospheric models, as well as for long-term assessment of model product uncertainties, particularly when applied to data from a global network of infrasound stations. A key aspect of this approach is that benchmarking between model and infrasound vespagrams considers all back-azimuth directions rather than just the direction of the dominant microbarom source, as done in several previous studies (Garcés et al., 2004; Hupe et al., 2019; De Carlo et al., 2019; Smirnov et al., 2021; De Carlo et al., 2021). The microbarom soundscape at a station is typically a sum of components stemming from a wide spatial distribution of ocean regions, and recently den Ouden et al. (2020) demonstrated that an iterative decomposition of the array spatial covariance matrix using the CLEAN algorithm (Högbom, 1974) can be exploited to resolve the back-azimuth and trace velocity of the most coherent wave front arrivals.

A long-term ambition is to exploit microbarom infrasound datasets to enhance the representation of stratospheric dynamics in atmospheric model products and hence increase the accuracy of both medium-range weather forecasting and sub-seasonal climate modeling (Büeler et al., 2020; Dorrington et al., 2020; Domeisen et al., 2020a, b). In addition to prospective numerical weather prediction improvements, the suggested vespagram-based approach may be applied in multi-technology studies of atmospheric dynamics, for example initiatives building on the Atmospheric dynamics Research InfraStructure in Europe (ARISE) projects (Blanc et al., 2018, 2019). These aim at harvesting from synergies between ground-based infrasound observations, radar and lidar systems, as well as airglow and satellite observations to monitoring the middle atmosphere (Chunchuzov et al., 2015; Le Pichon et al., 2015; Hupe et al., 2019; Smets et al., 2019; Hibbins et al., 2019; Assink et al., 2019; Le Pichon et al., 2019).

The study is organized as follows. The data and method are described in Sect. 2; the main results are presented in Sect. 3 followed by discussion in Sect. 4.

## 2 Materials and Methods

### 2.1 Infrasound dataset and signal processing

The infrasound array denoted IS37 or I37NO is located in Bardufoss, Norway (69.07° N, 18.61° E), and equipped with ten MB3 type (MB2005 prior to 2016) microbarometers over an aperture of 2 km (Figure 1a) (Fyen et al., 2014). This station is part of the International Monitoring System (IMS) which verifies compliance with the Comprehensive Nuclear-Test-Ban Treaty (CTBT) (Dahlman et al., 2009; Marty, 2019). The station was certified by the CTBT Organization on 19 December 2013 and is

operated by NORSAR, Kjeller, Norway (Schweitzer et al., 2021). Besides being included in the IMS, IS37 is also part of a regional network of European infrasound stations (Gibbons et al., 2007, 2015, 2019) that resolves significantly smaller events than the global IMS network (Le Pichon et al., 2008). In the framework of the regional network, data from IS37 has been used for multi-station studies characterizing European infrasound sources (e.g., Pilger et al., 2018).

The IS37 station routinely detects microbaroms within $0.1 - 0.6$ Hz originating from the North Atlantic, the Barents Sea, and beyond. An analytical expression for a plane-wave front incident on the IS37 array was used to characterize the array's integrated, frequency-dependent response in 0.1 Hz wide frequency bands from 0.1 to 0.6 Hz. The wave front was representative of a microbarom signal from the Atlantic Ocean, with a back-azimuth of $225°$ and an apparent velocity of 350 m/s typical of the stratospheric regime (Garcés et al., 1998; Whitaker and Mutschlecner, 2008; Nippress et al., 2014; Lonzaga, 2015). The base resolution of the array was taken to be the 1-sigma beam width of the Gaussian fitted to the array response at a constant velocity of 350 m/s (dashed line in Figure 1b) for each frequency band. The resulting resolution was found to be: $35°$, $23°$, $16°$, $13°$ and $10°$ for $0.1 - 0.2$ Hz, $0.2 - 0.3$ Hz, $0.3 - 0.4$ Hz, $0.4 - 0.5$ Hz and $0.5 - 0.6$ Hz band, respectively. It should be noted that this estimate is based on the homogeneous medium plane-wave time-delays between the array elements only and does not take into account meteorological conditions at the station, noise, or other coherence loss mechanisms that may result in a wider beam width.

In array signal processing, separating coherent from incoherent parts of the recorded signal, as well as the separation between different simultaneous arrivals are important concepts. When analyzing the wavefield in terms of a given horizontal slowness vector (e.g., described in terms of apparent velocity and back-azimuth), delay-and-sum beamforming (Ingate et al., 1985) is usually applied in combination with the underlying plane-wave model assumption. This method applies time-delays to the array sensor traces to focus on wave fronts arriving with a specific horizontal apparent velocity and a specific back-azimuth direction, hence amplifying wavefield components with the horizontal slowness of interest, while suppressing other components. However, the slowness vector models are not always accurate (Gibbons et al., 2020). In particular, the actual shape of the wave front arriving at infrasound arrays may differ from a theoretical plane-wave due to meteorological conditions and turbulence at the station, which make the underlying assumption of a locally homogeneous effective sound speed invalid. In this case, the beamforming is less efficient and the reduced array gain results in lower stack amplitude and signal distortion (Rost and Thomas, 2002).

To determine an unknown slowness vector component and to study the spatial structure of the wavefield over time, one can use the vespa (velocity spectral analysis) processing. This not only enhances the signal as the beamforming does, but also allows one to determine either the direction or apparent velocity of the incoming signal. The vespa method estimates the power of the signal either for a fixed apparent velocity with varying back-azimuth or for a fixed back-azimuth with varying apparent velocity. The result of the vespa processing is usually presented as an image displaying the power of incoming signal as a function of time and back-azimuth (or apparent velocity) called vespagram. Although the vespa is widely applied in seismological array data studies (e.g., Davies et al., 1971; Kanasewich et al., 1973; Muirhead and Datt, 1976; McFadden et al., 1986), it has not previously been exploited in peer-reviewed microbarom infrasound studies.

The vespa processing procedure described below is applied to each analyzed time window and frequency band:

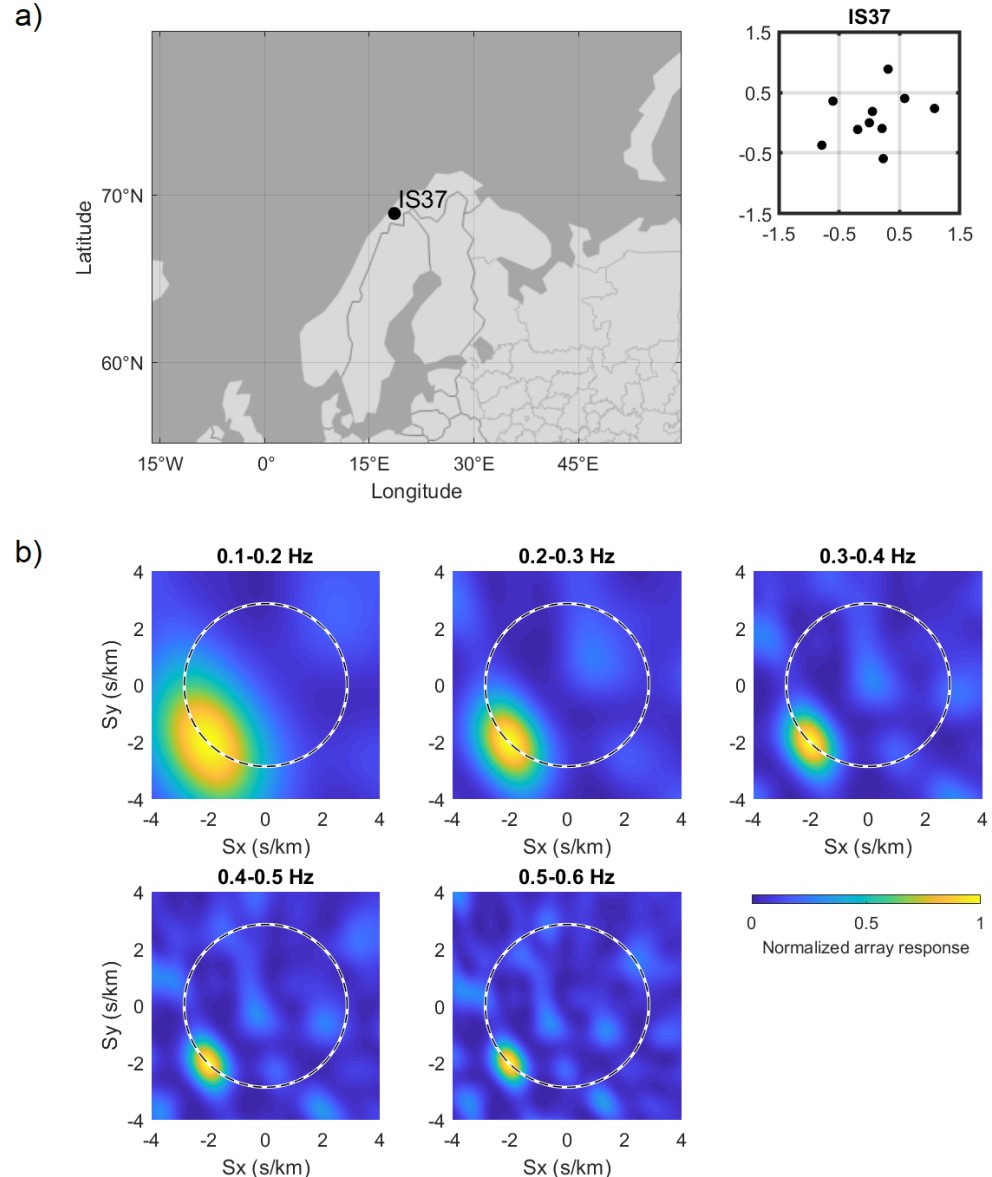

**Figure 1.** a) The IS37 infrasound array location and geometry. b) Integrated steered array response for 0.1 Hz wide frequency bands assuming a plane wave impinging at $225°$ back-azimuth and 350 m/s apparent velocity (indicated with a dashed circle). Here Sx and Sy denote the horizontal components of the slowness vector.

1) For each sensor $n$ of an array, we extract the signal trace $x_n(t)$ for the time window of interest. The analysis is done for an 1h moving time window, evaluated every 30 min. In general, the time series recorded at sensor $n$ at the location $\boldsymbol{r}_n$ can

be written as

$$x_n(t) = y(t - \boldsymbol{r}_n \cdot \boldsymbol{s}_{\mathrm{hor}}), \tag{1}$$

where $y(t)$ represents a plane wavefront, and $\boldsymbol{s}_{\mathrm{hor}}$ is the horizontal component of the slowness vector.

2) Remove the mean.

3) Apply a Butterworth bandpass filter to recordings. Calculations are performed for five equally spaced frequency bands that are within the microbarom frequency range (see Figure 1b).

4) Beam traces or delay-and-sum traces of an array with $N$ sensors are computed as

$$b(t) = \frac{1}{N} \sum_{n=1}^{N} x_n(t + \boldsymbol{r}_n \cdot \boldsymbol{s}_{\mathrm{hor}}). \tag{2}$$

In this study, classical linear vespa processing (Davies et al., 1971) is applied where the noise suppression is proportional to the square root of $N$ (Rost and Thomas, 2002). A beam is generated at each $1°$ in back-azimuth, for the fixed apparent velocity of 350 m/s. That allows to estimate signals coming from all directions but from approximately the same height corresponding to stratospheric altitudes.

5) Calculate the mean squared pressure (power) of each beam to get an estimate of the incoming signal strength as a function of back-azimuth and time.

Steps 1) – 5) are applied to all years of data analyzed. After the vespa processing, we apply a quality check based on the vespagram spectrum properties to exclude noisy data. At time windows when the vespa processing yields a directional spectrum with the power almost equal in all directions (the minimum exceeds 70 % of the maximum), data are ignored in our further analysis.

## 2.2 Microbarom source and propagation modeling

In this section we summarize the approach applied to get directional spectra of microbarom soundscapes as a function of time.

*Ocean wave model*. Hasselmann (1963) demonstrated that the source of microbaroms generated by non-linear interactions between counter-propagating ocean waves can be presented as an integral

$$H(f) = \int\limits_0^{2\pi} E(f_w, \theta) E(f_w, \theta + \pi) \, d\theta, \tag{3}$$

where $H(f)$ is the microbarom source spectrum, $E(f_w, \theta)$ is the power spectral density of the surface elevation, $\theta$ is the direction of the ocean wave propagation, $f_w$ is the wave frequency, $f$ is the acoustic frequency ($f_w = f/2$). Eq. (3) establishes a relationship between the microbarom source spectrum and the spectral densities of counter propagating waves for a given frequency and is called the Hasselmann integral. This is usually derived from the output of ocean wave models. This study uses the WAVEWATCH

III ® (The WAVEWATCH III® Development Group, 2016) ocean model. This model estimates $E(f_w, \theta)$ and its variations in space and time based on the surface wind fields. The model grid resolution is 0.5° in latitude and longitude and 3 h in time. The WWIII output includes many parameters based on $E(f_w, \theta)$. One of these parameters, namely $p2l(f_w)$, represents the spectral density of the equivalent surface pressure that force microbaroms (available at ftp://ftp.ifremer.fr/ifremer/ww3/HINDCAST/SISMO/, last access: 1 March 2021):

$$p2l(f_w = \frac{1}{2}\rho^2 g^2 H(f), \tag{4}$$

where $\rho$ is the water density, $g$ is the gravity acceleration. Hence, the Hasselmann integral needed for further calculations can be obtained from the WWIII model using Eq. (4). Studies on microseisms (e.g., Hillers et al., 2012) have demonstrated the limitations of a model that does not account for coastal reflection. These limitations have been accordingly raised in the context of microbaroms (Landès et al., 2014). Therefore, in this study the parametrization used to run the WWIII model accounts for fixed reflection coefficients of 10 % for the continents, 20 % for the islands and 40 % for ice sheets (Ardhuin et al., 2011).

*Microbarom source model.* A microbarom source model is basically a model transforming an ocean wave model output into acoustic radiation spectrum in the atmosphere. Here, calculations are based on the model by De Carlo et al. (2020), taking into consideration both the finite ocean-depth and the source radiation depending on elevation and azimuth angles. This microbarom model allows prediction of the location and intensity of the microbarom sources when applied to the Hasselmann integral. The Hasselmann integral is derived from the WWIII model. The output of this step is an acoustic spectrum for each cell of the wave model.

*Microbarom propagation in the atmosphere.* The next step is to account for the atmospheric influence on the microbarom propagation and ducting. For example, 3-D ray tracing or full-waveform approaches would provide a more accurate simulation of the infrasound propagation, but these methods involve a larger computational burden (De Carlo et al., 2021). Instead, the semi-empirical attenuation law by Le Pichon et al. (2012) is used in the current study. This law is applied to the microbarom source spectra obtained in the previous step. This law accounts for the distance between the source and the station as well as for the frequency but assumes a horizontally homogeneous atmosphere. The atmospheric conditions at the station location are taken into account via $V_{\text{eff-ratio}}$, which is the ratio of the effective sound speed in the propagation direction at 50 km altitude and the effective sound speed in the same direction on the ground. The atmospheric wind and temperature needed to determine the $V_{\text{eff-ratio}}$ are derived from the European Center for Medium-range Weather Forecasts (ECMWF) High Resolution (HRES) model (http://www.ecmwf.int). The atmospheric wind and temperature at the station location are extracted from a $0.5° \times 0.5°$ model grid using bilinear interpolation. The temporal resolution of the ECMWF HRES is 6 h, and the assumption of the constant atmospheric wind and temperature over this time period is made to avoid possible discrepancy caused by interpolation in time. The output of this step is acoustic spectra attenuated to reflect what would be seen by the station.

*Summation of sources.* To obtain the directional spectrum at the station, all attenuated spectra from model cells within 1° azimuth band and less than 5000 km away from the station are summed. The distance limitation comes from the attenuation law definition. Although this attenuation law is widely used for propagation over very long distances (Smirnov et al., 2021; Pilger et al., 2019; Hupe et al., 2019; De Carlo et al., 2019), it was designed for distances up to 3000 km only. The choice of the

maximum distance depends on the location of the station and the main sources, as well as on how realistic spectrum is needed for a specific task. Recently, De Carlo et al. (2021) provided a multi-station comparison between PMCC-processed microbarom data and the microbarom model by De Carlo et al. (2020) which is also used in the current study. Results for 45 IMS stations in (De Carlo et al., 2021) demonstrate that integrating microbarom sources at distances up to 5000 km provides realistic spectra. Thus, all sources that are more than 5000 km away from the IS37 station are excluded from this study.

After applying these steps, and integrating over the frequency bands, we get an estimate of the microbarom power spectral density as a function of time and back-azimuth, just as vespagrams. However, vespagrams cannot be directly compared to the modelled microbarom soundscapes since the latter do not take into account the frequency-dependent array resolution. Therefore, we smooth the modelled microbarom soundscapes by convolving with a Gaussian kernel at each time step taking into account cyclical nature of back-azimuth when smoothing near $360°/0°$. Kernels are normalized to unit area, and their standard deviations (width) decrease with frequency (see Sect. 2.1).

## 2.3 Similarity index

This section introduces an approach to benchmark the microbarom model against the infrasound data. Both datasets need to be of the same temporal resolution to assess a similarity at each time step. In the current study, in order to avoid interpolating the model output in time, the vespa processing output is sub-sampled to match the three-hourly microbarom model grid. Further, all results are presented with a temporal resolution of 3 h.

Figure 2 presents a difference in the direction of the maximum power between the vespagram and either the model or the smoothed model. Both medians and uncertainty ranges are estimated based on the back-azimuth difference at the maximum power only. Uncertainty values falling into the 25 to 75 percentile range are an objective assessment of the discrepancy between the model and vespagrams. These values originate from the wintertime when atmospheric conditions are favorable for the eastward ducting (Sect. 3.1). In summer, atmospheric conditions are not so stable and there are several factors that can cause model-vespagram discrepancies (Sect. 3.1) accompanied by an increase in the uncertainty range. Note that after the smoothing (Sect. 2.2), there is a better agreement between the model and the vespagram leading to a decrease in the median and the uncertainty ranges (Figure 2).

A similarity index (SI), inspired by comparison approaches in image processing, is introduced as

$$\text{SI}(t) = 1 - \text{MSE}(t) = 1 - \frac{1}{N_\theta} \sum_\theta \left[ P_{\text{model}}(t, \theta) - P_{\text{vespa}}(t, \theta) \right]^2, \tag{5}$$

where MSE is the mean squared error (or difference) between the normalized smoothed model output, $P_{\text{model}}(t, \theta)$, and the normalized vespagram, $P_{\text{vespa}}(t, \theta)$, calculated at each time step, where $\theta$ is back-azimuth and $t$ is time. This SI metric is hence insensitive to the total microbarom power but instead provides information on how accurate the model reproduces the directional pressure spectrum in the recorded data. SI equal to one indicates a full match between model and infrasound vespagram in terms of the back-azimuthal power distribution.

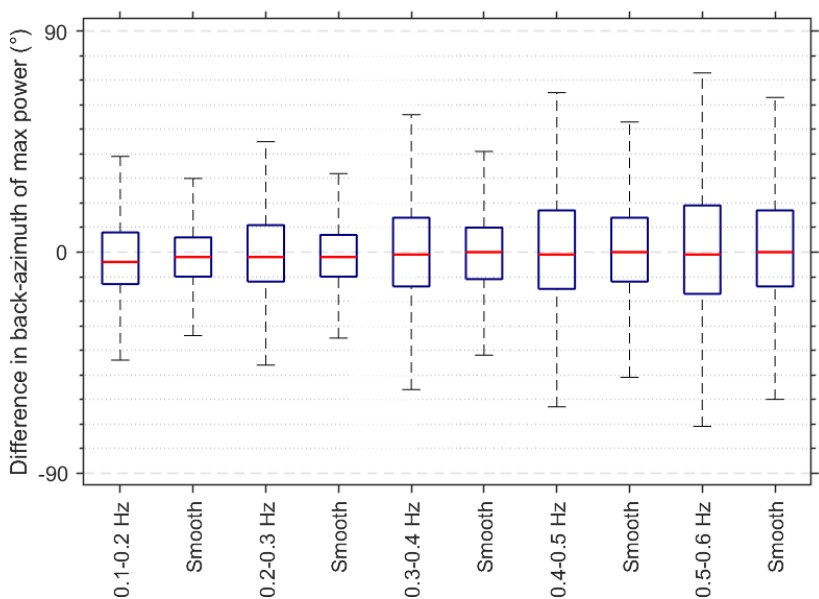

**Figure 2.** Difference in the direction of the maximum power between i) model and vespagram (indicated with a frequency band name in $x$-axis) and ii) smoothed model and vespagram (denoted as "smooth" in $x$-axis) over 6 years of data. The red lines present the median, blue boxes indicate the 25 to 75 percentile range, and whiskers correspond to $\pm 3$ standard deviations.

## 3 Results

### 3.1 Comparison for full seasons

This section presents a multi-year model-vespagram comparison, focusing on microbarom characteristics over different seasons. We start with a detailed look at 2016 results, followed by an analysis of all 6 years.

Figures 3 and 4 present benchmarking microbarom model and vespa processing images (vespagrams) for two frequency bands, namely $0.1 - 0.2$ Hz and $0.5 - 0.6$, for 2016. These figures contain 8 panels each, which are discussed in more details below. Figures 3a and 4a show the maximum amplitude per time step over one year, i.e. the dominant signals in the azimuthal spectra. Enhanced ocean source activity during winter is accompanied by the eastward stratospheric wind favorable for ducting infrasound over long distances (Le Pichon et al., 2006). This results in a maximum of microbarom pressure amplitude both in model and vespagrams regardless of frequency band. The microbarom radiation model by De Carlo et al. (2020) accompanied by the semi-empirical wave attenuation law (Le Pichon et al., 2012) generally reproduces infrasound amplitudes accurately. An exception is between day 200 and 210 in 2016 (Figure 3a), with the modelled amplitude much lower than the amplitude obtained via the vespa processing. This discrepancy may indicate an overestimation of the attenuation caused by errors in the atmospheric model wind due to the range-independent simplification.

Figures 3b – d and 4b – d display microbarom soundscapes as predicted by the model, smoothed model and vespa processed recordings, respectively. Here the soundscapes are plotted in terms of the base-10 logarithm of the amplitude, in order to allow for showing the weaker summertime microbarom amplitudes in the same display as the stronger wintertime signals. Note that the vespagrams are noisier during summer (grey fields in Figures 3g and 4g) especially for the $0.1 - 0.2$ Hz band.

Due to the strong seasonal variability of the microbarom amplitude, it is difficult to compare the direction of winter to summer detections on an absolute amplitude scale. Thus, we normalize Figures 3b – d and 4b – d by the maximum amplitude at each time step (see Figures 3e – g and 4e – g, right) and estimate the directional distribution of the dominant signal in $10°$ bins (see Figures 3e – g and 4e – g, left). For the frequency band of $0.1 - 0.2$ Hz the North Atlantic is the dominant source direction throughout the year (the main peak in Figure 3e – g, left). However, the maximum power from north-easterly and south-easterly

directions sometimes also observed in summer. To generate infrasound at such low frequencies, the source needs to be of a substantial spatial extent, and therefore there is a limited number of possible oceanic sources. After comparison with the model maps, we interpret these arrivals as microbaroms generated in the Pacific and Indian Oceans, respectively. The stratospheric summertime westward wind could guide the infrasound waves towards the IS37 station. Unlike what is seen in the vespa processed recordings, the model doesn't predict microbaroms originating from the Indian Ocean direction. A plausible reason

for this is that the distance between the station and the Indian Ocean source region is much greater than the maximum distance of 5000 km included in the modelling (see Sect. 2.2). Looking at higher frequencies, there is a pronounced change in the dominant direction of the source from the Atlantic in winter to the Barents Sea and the Greenland Sea in summer (peaks in Figure 4e–g, left). This is associated with the change of wind direction in the stratosphere from eastward to westward. Analysis of 6 years dataset in terms of the dominant source direction indicates three prevailing microbarom source regions associated with the

North Atlantic, the Greenland Sea, and the Barents Sea. These appear at the vespagram (model) back-azimuths of $266° \pm 14°$ $(265° \pm 16°)$, $339° \pm 14°$ $(345° \pm 8°)$ and $26° \pm 6°$ $(34° \pm 7°)$.

    Figures 3h and 4h present values of SI obtained over a year. In winter, SI for lower frequencies is stable and has values $\sim 1$, with exceptions corresponding to increased noise level in vespagrams or to SSW events that are discussed in Sect. 3.2. Relatively low SI for higher frequencies can be explained either by spurious apparent sources corresponding to array response function

side-lobes (Figure 1b) or by the presence of local sources in the vespagram that are missed or not-well reproduced in the model. In summer, SI values are quite variable and unstable but never fall below $0.5$. Such behavior is typical regardless of year and frequency band (see Figure 5 for a multi-year comparison). One possible explanation is the changing weather conditions present at the station throughout the year. For example, Orsolini and Sorteberg (2009) have shown an enhancement in the number and intensity of summer cyclones in the Arctic and Northern Eurasia. This would result in additional wind and rain noise in the

infrasound recordings that would especially be enhanced at the lower frequencies. Another possible contribution would be the poor resolution of the array at low frequencies that can mix stratospheric signals with those from higher altitudes. These sometimes dominate at IS37 in summer (Näsholm et al., 2020) but are not included in the model. The relative stability of the model's results in Figure 4e –f relative to the vespagram would indicate that there are additional sources of variability, either atmospheric, source region, or propagation path, that are not well characterized in the model.

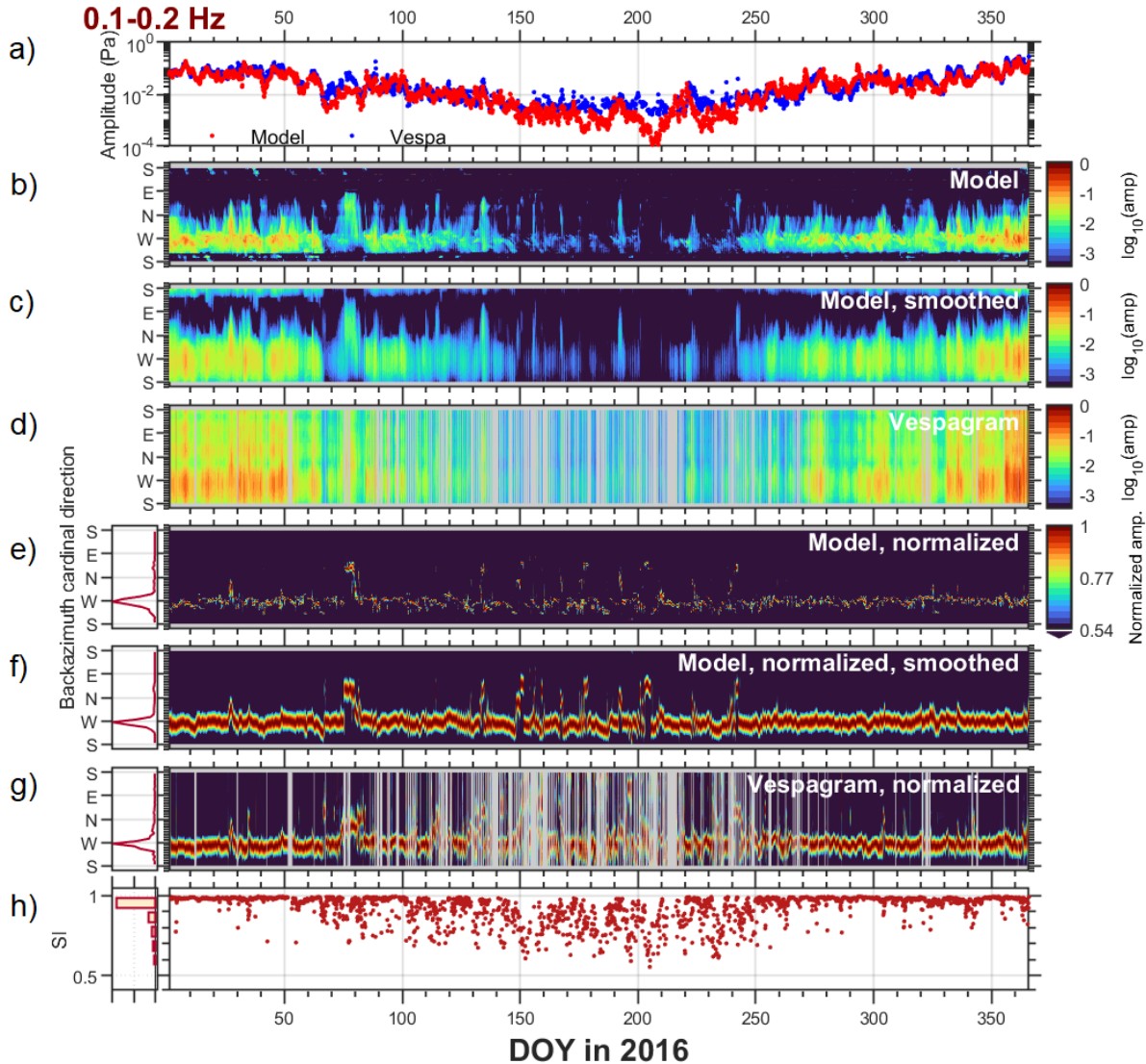

**Figure 3.** Benchmarking microbarom model and infrasound vespagram for 0.1 – 0.2 Hz in 2016 for the IS37 station. a) amplitude of the dominant signal (blue – vespa processing, red – model); b) the base-10 logarithm of the model amplitude; c) the base-10 logarithm of the smoothed model amplitude (Sect. 2.2); d) the base-10 logarithm of the infrasound vespagram amplitude (Sect. 2.1); e) – g) (right) same as panels b) – d) but after normalization by the maximum amplitude at each time step; e) – g) (left) normalized directional distribution of the dominant signal (10° bins); h) similarity index between panels f) and g) (right) and its normalized distribution (left). Gray fields indicate periods where infrasound data are disregarded due to noise and an indistinct directional spectrum. Panels b) – g) are visualized using the Turbo colormap (Mikhailov, 2019).

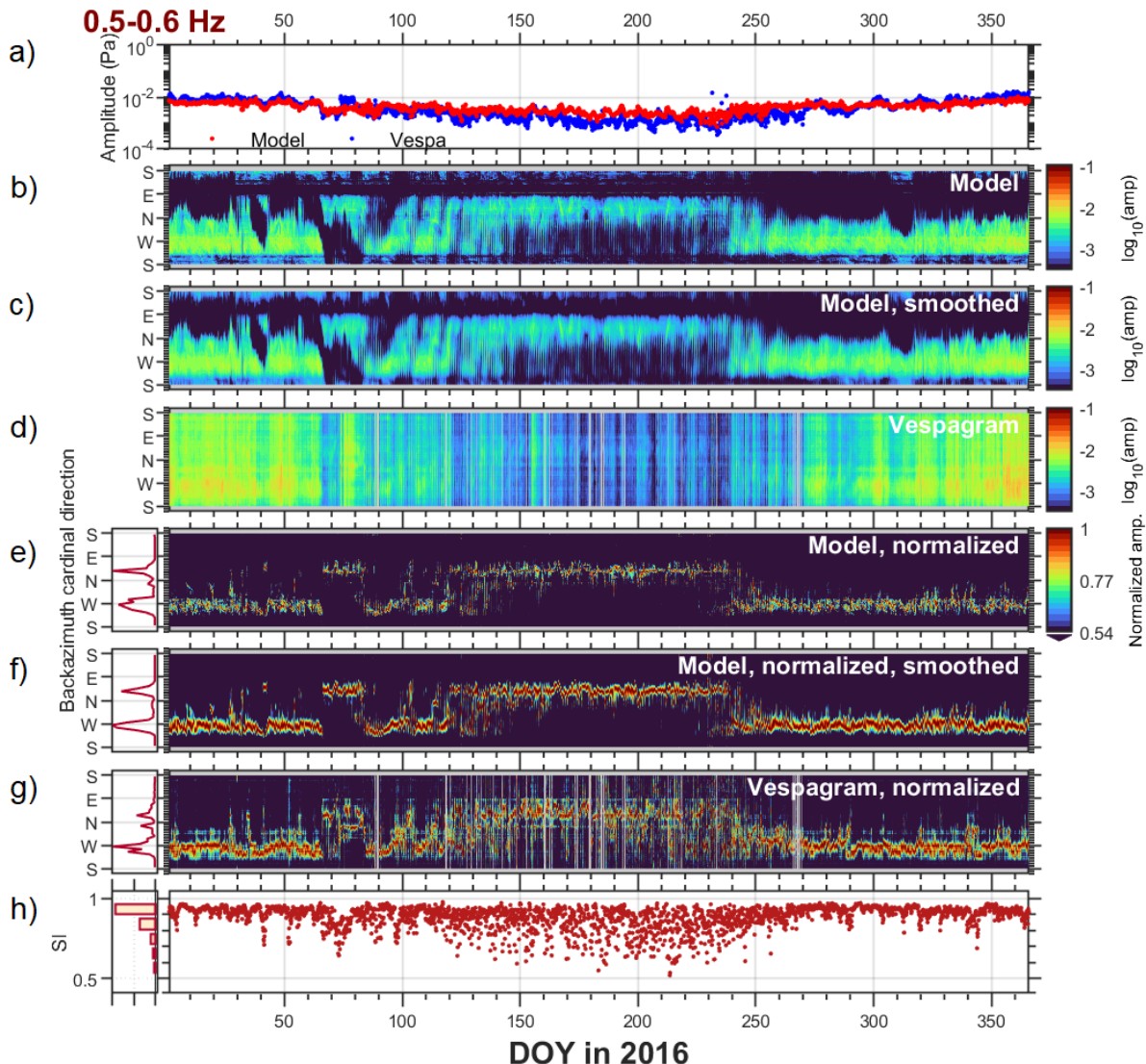

**Figure 4.** Same as Figure 3 but for 0.5 – 0.6 Hz.

As indicated by high SI values, especially in winter, the infrasound data processed in the framework of the vespa approach are in a good agreement with modelled microbarom soundscapes in both time (seasonal variations) and space (directional distribution). The similarity estimation proposed allows detection of inconsistencies between the microbarom model and the vespa processing which might be used for identifying biases in atmospheric models. This is especially promising for low frequencies where side-lobes of the array response do not appreciably affect the analysis.

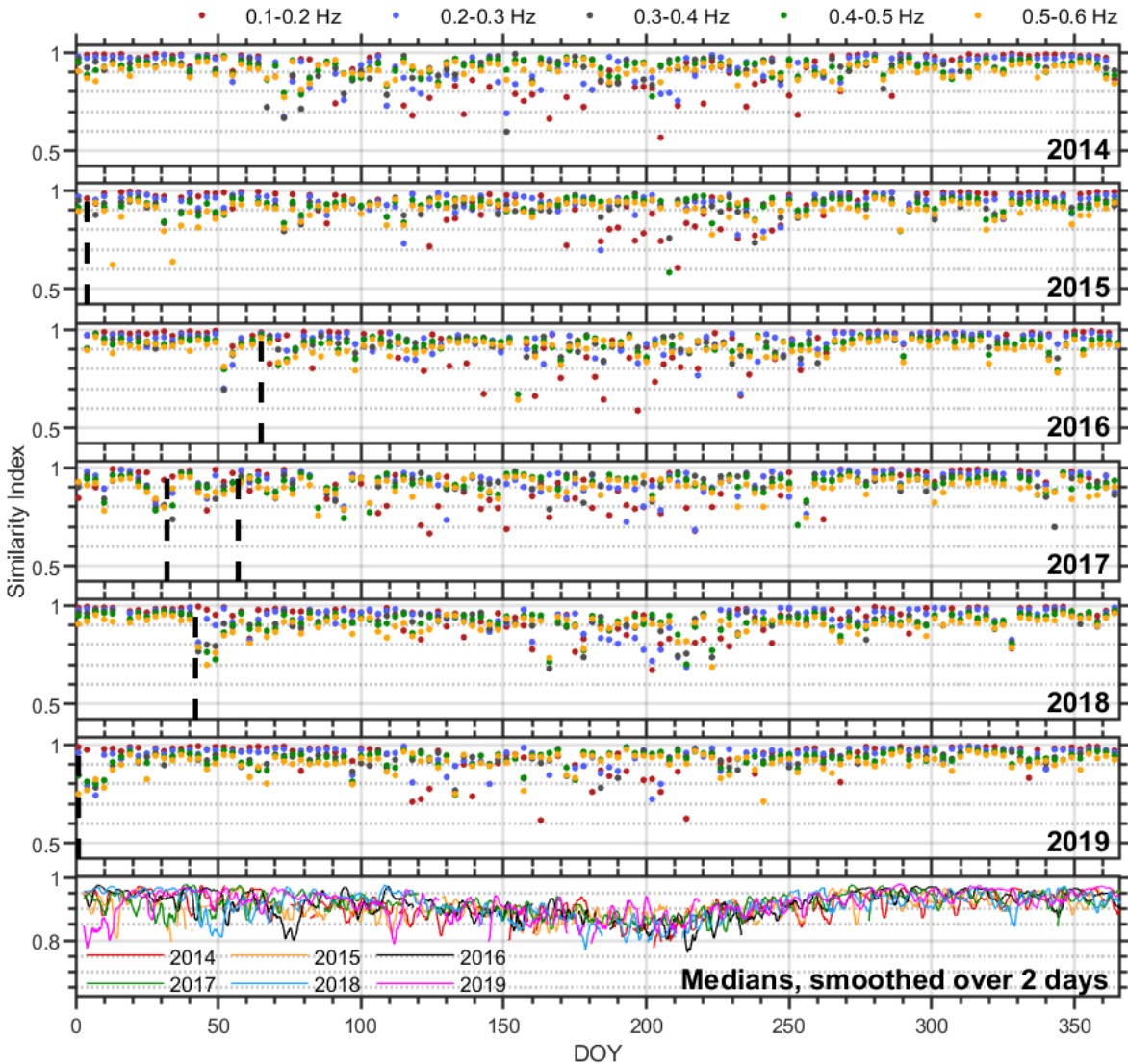

**Figure 5.** Multi-year comparison between vespagrams and smoothed modelled microbarom soundscapes at the IS37 station. The similarity index is color-coded depending on frequency band: $0.1-0.2$ Hz – red, $0.2-0.3$ Hz – blue, $0.3-0.4$ Hz – grey, $0.4-0.5$ Hz – green, $0.5-0.6$ Hz – orange. Data are presented as a discrete set with a time step of 3-days (day 1 00 h, day 4 00 h etc.). Black dashed lines present SSWs onsets. Medians over frequency bands in the last panel are color-coded depending on year: 2014 – red, 2015 – orange, 2016 – black, 2017 – green, 2018 – blue, 2019 – magenta.

## 3.2 Examination of major sudden stratospheric warmings

Although this is not the main objective of the current study, in this section we examine the ability of the vespagrams to detect extreme atmospheric events, such as sudden stratospheric warmings (SSWs), and compare the model and the vespa processing for six selected events.

SSWs usually occur in wintertime and are, in general, associated with a sudden and short increase in stratospheric temperature and mesospheric cooling at high and middle latitudes (Shepherd et al., 2014; Butler et al., 2015; Limpasuvan et al., 2016; Zülicke et al., 2018). SSWs are often classified into minor and major warmings, depending on whether there was a weakening or reversal of the zonal wind (Butler et al., 2015). During the period of our consideration, three major and three minor SSWs occurred. Major SSWs took place with onsets on 5-6 March 2016 (Manney and Lawrence, 2016), 11 February 2018 (Rao et al., 2018; Lü et al., 2020) and 1 January 2019 (Rao et al., 2019, 2020), while the minor events occurred with onsets on 4 January 2015 (Manney et al., 2015; Mitnik et al., 2018), 1 and 26 February 2017 (Eswaraiah et al., 2020). Note that there can be an error up to several days in determining SSW onset day since there is no single way to define the onset, and different authors use different definitions. A prime example is the first SSW in 2017. According to the definition of the World Meteorological Organization, this event is classified as minor, but in a number of studies it is referred to as major (Xiong et al., 2018; Conte et al., 2019). Vertical dashed lines in Figures 5 – 6 correspond to the onset days listed above when SSW criteria were met.

The infrasound signature reported by Donn and Rind (1971) and Evers and Siegmund (2009), which showed a significant change in direction of the infrasound arrival due to a change in favorable stratospheric waveguide, can be seen in Figure 6 for all SSWs under consideration and in Figures 3e – g and 4e – g for the 2016 SSW. The change in direction from the North Atlantic to the Barents Sea is clearly pronounced in both model and vespagrams around SSWs onset days. Figure 3f – g demonstrates that the signature appears late in the model data and its duration is much shorter than in the vespagram, analogous to study by Smets et al. (2016). For higher frequencies (Figure 4f – g) the duration of a change from eastward to westward pattern is longer and continues until late March or early April, which corresponds to reanalysis data (Manney and Lawrence, 2016).

Another feature revealed is a significant decrease in similarity index between the model and vespagrams during SSWs (Figure 5) which is characteristic for all events under consideration. The smallest discrepancies in the direction of the dominant wave front between the model and infrasound data during SSWs reach about $5° − 7°$, but the largest reach as much as $90° − 100°$ (Figure 6). This may be caused by the following factors. In most cases, the back-azimuth change around SSWs onsets appears earlier in the vespagrams than in the model with the difference of 3 to 24 h. Note that this also depends on the frequency band. Similar results were previously obtained by Smets and Evers (2014) and can be explained by the presence of an error in determining a SSW onset day from reanalysis data because of a scarcity of observations at stratospheric altitudes (Charlton-Perez et al., 2013) or by inadequate stratospheric analysis and forecast during SSWs as addressed by Diamantakis (2014) and Smets et al. (2016). Sometimes the SSW signature does not appear in the vespagram while appearing in the model (see Figure 6 around SSW 2018 onset day for example). This can arise when employing a horizontally homogeneous atmosphere and overly constraining the model with the ECMWF wind and temperature at 50 km altitude. Such approach does not allow a full, altitude dependent description of infrasonic waves in the atmosphere and causes discrepancies between model and vespagrams.

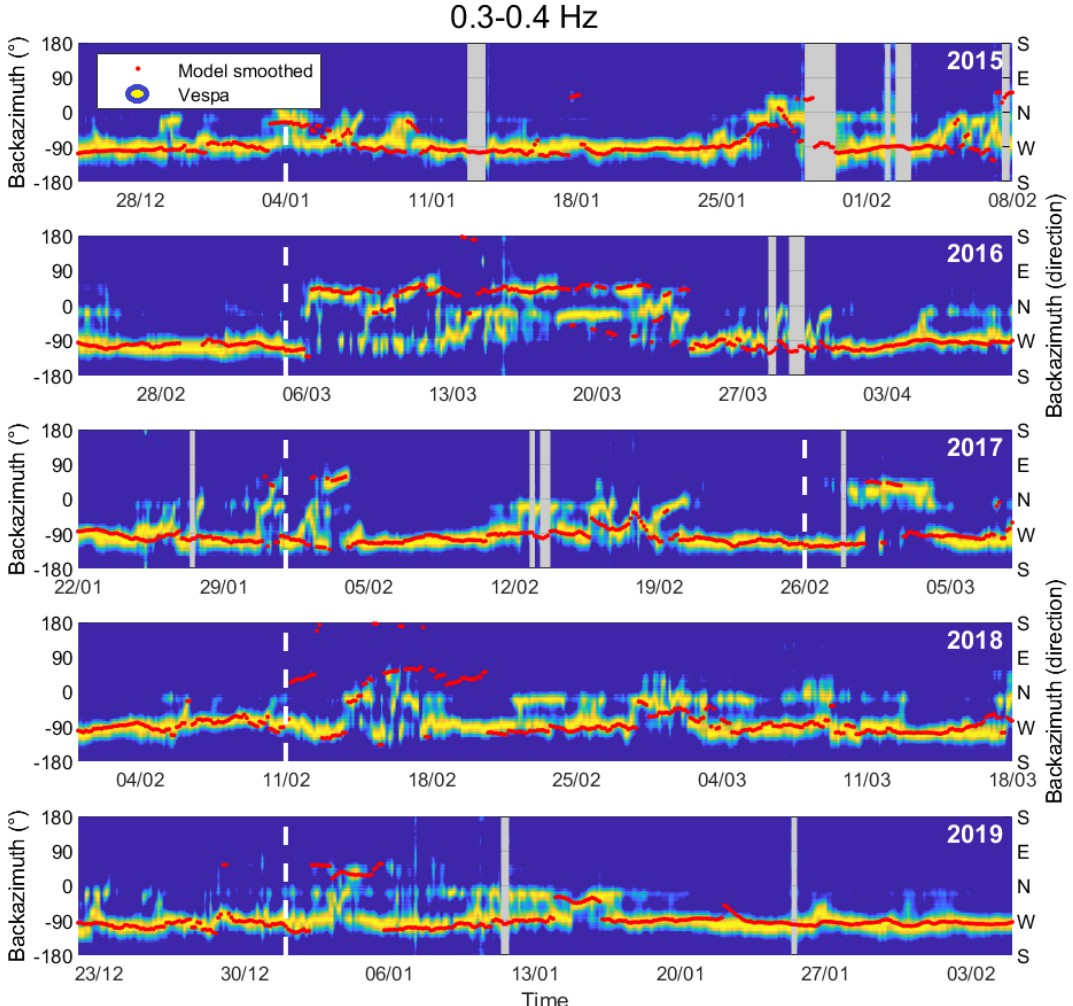

**Figure 6.** Comparison between microbarom azimuthal distributions at IS37 (vespagrams) normalized per time step, and the back-azimuths of the dominant signals as predicted by the model (red dots). The results are shown around SSWs $2015 - 2019$ for the $0.3 - 0.4$ Hz band. White vertical dashed lines indicate onset days when SSWs (minor or major) criteria were met. Gray fields indicate periods where infrasound data are disregarded due to noise and an indistinct directional spectrum.

Considering long propagation path for microbaroms, net wind effect along the propagation path can be equal to zero in the vespagram in contrast to the model, which estimates the probability of the signal arrival at the final point of the path. It has been demonstrated by Evers and Siegmund (2009) and Smets and Evers (2014) that the ECMWF wind direction not always characterize the actual infrasound path, resulting in the above-mentioned model-vespagram discrepancies.

Despite slight difference in the dominant direction of the wave front arrival during SSW events, both model and vespagrams reproduce changes in the infrasound pattern correctly in time. Moreover, since vespagrams can detect changes in the stratospheric dynamics during extreme events, there is a potential in using it for near-real-time stratospheric diagnostics.

## 4 Discussion and Conclusions

In this study, we compare observed and predicted microbaroms soundscapes using a vespagram-based approach. Analysis is performed based on calculation of microbaroms power as a function of time and back-azimuth at a constant apparent velocity of 350 m/s. Note, however, that the vespagram-family of time-dependent microbarom data visualizations can be constructed also using other array processing techniques that estimate power as a function of the slowness of the wave front, e.g., using robust estimators as explored by Bishop et al. (2020), or adaptive high-resolution approaches like Capon's method (Capon, 1969). An advantage of the vespagram-approach is that microbarom radiation and propagation models can be benchmarked against recorded infrasound data for all directions simultaneously, as opposed to methods where only the back-azimuth direction of maximum power is considered (e.g., Hupe et al., 2019; Smirnov et al., 2021). Since the vespa processing is computationally low-cost and able to track variations in microbarom parameters over extended periods spanning one or several years, it can be utilized for near-real-time assessment of atmospheric model products and for developing infrasound-based stratospheric diagnostics. It can also be used when assessing changes in infrasound signatures over shorter time windows, e.g., during extreme atmospheric events.

Limitations in this study are predominantly related to microbarom propagation modelling. In addition to the scarcity of observations at the stratospheric altitudes (Charlton-Perez et al., 2013) which affect the accuracy of directional distribution of predicted microbarom soundscapes, the horizontally homogeneous atmospheric approximation used in the study creates substantial limitations. In fact, the atmosphere is not homogeneous. It has horizontal and vertical inhomogeneities in the atmospheric wind and temperature fields caused, for example by gravity waves, tides and SSWs. These atmospheric irregularities significantly affect the long-distance infrasound propagation and, as a result, the directional spectra detected at the reception point. Moreover, the modelling would benefit from applying a full-waveform simulation code for the propagation of the radiated microbaroms to the station (e.g., Assink et al., 2014; Kim and Rodgers, 2017; Brissaud et al., 2017; Petersson and Sjögreen, 2018; Sabatini et al., 2019). This would provide a more refined modelling of the atmospheric ducting compared to the semi-empirical approach (Le Pichon et al., 2012) applied in the current study. An alternative which is less computational expensive is (3-D) ray-tracing, which can account for both range-dependent atmospheric models and cross-wind effects (e.g., Smets and Evers, 2014; Smets et al., 2016). However, the inherent high-frequency approximation of the ray-theory can limit the modelling of diffraction and scattering effects (Chunchuzov et al., 2015) that can be important for the low-frequency microbaroms. Also note that more advanced simulations of infrasound propagation would require wind and temperature profiles with a high vertical resolution (or an appropriate stochastic parametrization) to account for the effect of small-scale atmospheric irregularities on microbarom scattering. Resolving small-scale structures in atmospheric models, reanalysis and forecasting systems remains a topic of active research. Several research efforts were made to develop methods exploiting infrasound observations to improve

the representation of wind and temperature in atmospheric model products (e.g., Chunchuzov et al., 2015; Assink et al., 2019; Amezcua et al., 2020; Vera Rodriguez et al., 2020).

A more elaborate microbarom propagation model could also allow for an estimate of the full microbarom wavefield impinging an infrasound station, hence providing an estimate of its power within the full horizontal slowness space of plane wave front directions (or a selected relevant region). This way, we could benchmark modelled and recorded microbarom fields at an infrasound array for each sliding time window in the full horizontal slowness domain, without restricting the analysis to the region around a fixed apparent velocity as carried out in the current study. Notably, such "slowness plots" of modelled and recorded microbarom power are also (time-varying) images which can be assessed and compared using the versatile ecosystem of image processing and image comparison algorithms.

Future developments can include compilation of long-term time-dependent statistics of similarity between model and infrasound recordings for multiple stations on global and regional scales. This would allow the definition of anomaly flag criteria which would indicate unexpected inconsistencies between model and observations due to, for example, biases in atmospheric model products. Moreover, we suggest applying the approach presented here to global assessment and comparisons of ocean wave-action model products, as well as to validation and further refinement of microbarom radiation estimation algorithms.

 **Appendix A: Apparent velocity statistics**

Figure A1 displays histograms of the apparent velocity detection statistics calculated using Progressive Multi-Channel Correlation (PMCC) processing (Cansi, 1995) for all frequency bands and years. These support the choice of 350 m/s as apparent velocity in the vespagram calculations for the analysis of microbaroms ducted through the stratosphere.

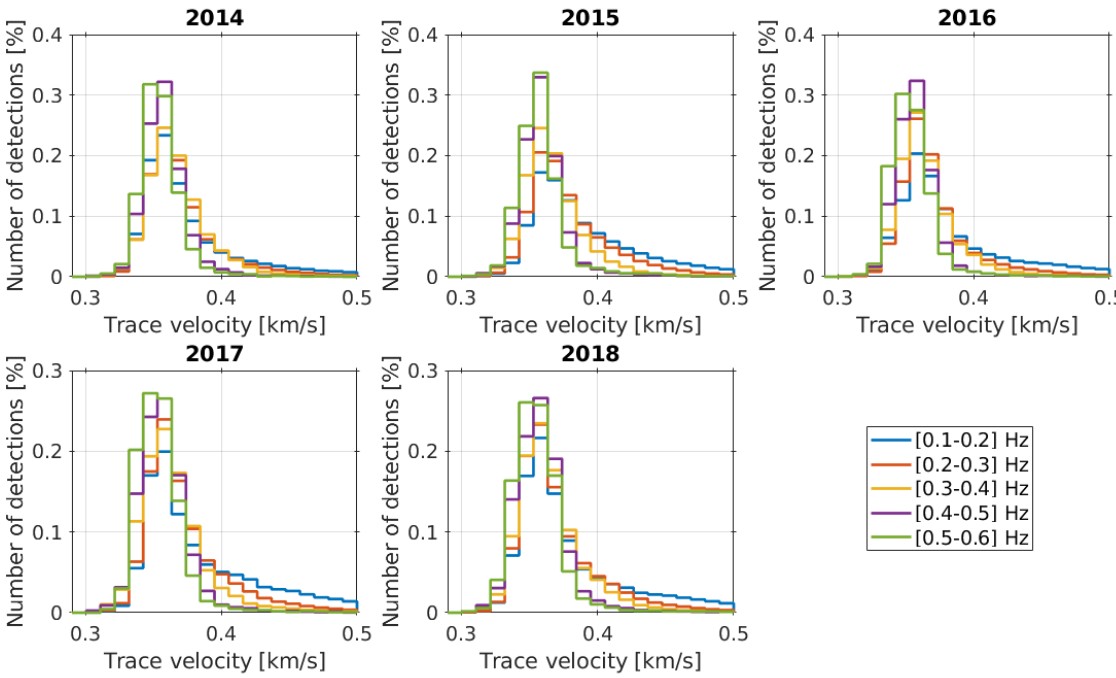

**Figure A1.** Trace (apparent) velocity values as predicted by the PMCC for the IS37 infrasound station in 2014 - 2018. Frequency bands are color-coded: blue $-0.1-0.2$ Hz, orange $-0.2-0.3$ Hz, yellow $-0.3-0.4$ Hz, purple $-0.4-0.5$ Hz, green $-0.5-0.6$ Hz.

*Author contributions.* EV and SPN developed the vespagram calculation code, performed the infrasound data processing, and made the model/data analysis. MDC and ALP developed the microbarom model code and performed the simulations. EV prepared the manuscript with contributions from all co-authors. PJE supervised the work and reviewed the results. SPN and ALP initiated the study.

*Competing interests.* The authors declare that they have no conflict of interest.

*Acknowledgements.* This study was facilitated by previous research performed within the framework of the ARISE and ARISE2 projects (Blanc et al., 2018, 2019), funded by the European Commission FP7 and Horizon 2020 programmes (Grant agreements 284387 and 653980). The authors are thankful to Igor Chunchuzov and one anonymous reviewer for valuable suggestions and constructive review of the paper. EV and SPN are also grateful to Y. J. Orsolini for discussion and valuable comments.

*Financial support.* This research was supported by the Research Council of Norway FRIPRO/FRINATEK basic research programme, project contract 274377: *Middle Atmosphere Dynamics: Exploiting Infrasound Using a Multidisciplinary Approach at High Latitudes* (MADEIRA).

*Review statement.* To be determined.

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
