# Peer review of "Microbarom Benchmarking microbarom radiation and propagation model assessment using against infrasound recordings: a vespagram-based approach"

_Annales Geophysicae, 2020_

## Referee Comment (RC1) · Igor Chunchuzov (Referee) · 4 Jan 2021

Review of the paper by Vorobeva et al. "Microbarom radiation. . .."

This paper compares the results of the analysis (called vespagram-analysis) of the time dependencies for the time period from 2011 to 2016 of the amplitude, apparent horizontal velocity and azimuth of microbaroms (infrasound with frequencies of 0.1-0.6 Hz , which is assumed to be generated by the interacting counter-propagating sea surface waves) with the results of modeling of the generation and propagation of microbaroms. The comparison showed that the model used, in general, predicts well the temporal variations in the amplitude and azimuth of arrival of the stratospheric reflections of the mirobaroms with a fixed horizontal apparent velocity of 350 m/s. These results certainly deserve publication in the journal Annales Geophysicae. The paper shows that a vespagram-analysis enhances a signal recorded by an array of infrasound sensors at the infrasound station IS37 (Norway), similarly to that as the beamforming method does, but it also enables to determine the azimuth or apparent horizontal velocity of the signal as a function of time. With this method, it was possible to conduct continuous monitoring of seasonal changes and sudden stratospheric warmings in the temporal variations of the microbarom amplitude and azimuth. I found a number of minor remarks (below) which should be taken into account when revising this paper. The microbarom model used is based on microbarom generation model that predicts the spatial distribution of the acoustic sources over the ocean surface, and on the atmospheric model that allows one to calculate the microbarom propagation from the microbarom sources to the receivers. Each of these models has its own drawbacks, which introduce errors in the prediction of the parameters of microbaroms at distances of thousands of kilometers from their sources. One of the drawbacks of the propagation model, which the authors themselves pointed out, is the approximation of a horizontally homogeneous atmosphere. The presence in the real atmosphere of the horizontal inhomogeneities in the wind velocity and temperature significantly affects the azimuth of arrival of the signal at the reception point and the prediction of the source back-azimuth. 2) Another disadvantage of the propagation model used is that the wind velocity and temperature profiles derived from the European Center for Medium range Weather Forecasting (ECMWF) do not have sufficient vertical resolution to account for the effect of small-scale atmospheric irregularities on microbarom scattering and, as a result, on amplitude attenuation with increasing distance from a source for different directions of propagation. 3) When describing microbarom generation model the authors refer to the state-of-the-art microbarom radiation theory (De Carlo et al., 2020), which "…allows prediction of the location and intensity of the microbarom sources when applied to the Hasselmann integral." It would be important to note briefly in the paper of how are the frequency spectra of counter propagating waves derived in the wave model

to calculate the Hasselmann integral, because the latter defines the distribution of the intensity of acoustic sources over ocean surface. 4) The parameters of microbaroms vs time were obtained for the fixed apparent velocity of 350 m/s, which corresponds to the arrivals of the signals from the stratospheric altitudes. Are there in the detected signal the microbarom reflections from the lower thermosphere with another apparent velocity? 5) The amplitude obtained from the model in Fig.2a (red) in the time interval 200-201 DOY is two orders lower than the amplitude obtained by vespa processing. Could you explain the cause of such discrepancy? 6) Figure 2f and Figure 2j: Are the amplitudes (model and vespagram) in these figures normalized by the maximum amplitude? 7) The last expression in the right side of (3) defines rather a mean squared error (MSE), than a similarity index (SI), since this expression becomes zero (not 1) in case of full match between model and infrasound vespagram. 8)Line 185: "Going to higher frequencies, there is a pronounced change in the dominant direction of the source from the Atlantic in winter to the Barents Sea in summer (Figure 3)."

Do the higher frequencies react stronger on the change of wind direction in the stratosphere from eastward to westward than the lower ones? If yes, then why?

Igor Chunchuzov

Please also note the supplement to this comment:
https://angeo.copernicus.org/preprints/angeo-2020-78/angeo-2020-78-RC1-supplement.pdf

---

## Referee Comment (RC2) · Anonymous Referee #2 · 7 Jan 2021

This study presents a novel method for analyzing infrasound data, the vespagram approach. The authors apply this method to 6 years of data recorded at the Norwegian array IS37 to extract microbarom arrivals. The latter are compared with simulations using a recently developed microbarom model. For comparison, the similarity index is introduced, which is based on mean-squared errors. While the manuscript title partly raises expectations on conclusions about the microbarom model performance, the manuscript essentially evaluates the vespagram approach as a method to utilize infrasound for stratospheric diagnostics. Due to its capability to scan all directions simultaneously, the advantage of this novel approach is obvious. The manuscript is generally well written and organized, but the presentation and discussion of the results need to be

enhanced. In particular, Section 3.1 confronts the reader with three extensive figures within a few sentences while remaining sparse with explaining and discussing details (e.g., on remarkable features, outliers) of these figures. For instance, Figure 4 is only once briefly referred to. My specific comments indicate several features that could or should be further discussed (e.g., Figs. 2 and 3). Given the number of comments and questions, I request a major revision, rather than a minor. Once the authors have addressed these comments and questions, this study will be an important contribution and thus certainly worthy of publication in Annales Geophysicae.

The specific comments and questions, and a few technical suggestions, are included in the supplement.

Please also note the supplement to this comment:
https://angeo.copernicus.org/preprints/angeo-2020-78/angeo-2020-78-RC2-supplement.pdf

**Supplement:**

Referee comment: "Microbarom radiation and propagation model assessment using infrasound recordings: a vespagram-based approach" (Vorobeva et al.)

General comments

This study presents a novel method for analyzing infrasound data, the vespagram approach. The authors apply this method to 6 years of data recorded at the Norwegian array IS37 to extract microbarom arrivals. The latter are compared with simulations using a recently developed microbarom model. For comparison, the similarity index is introduced, which is based on mean-squared errors. While the manuscript title partly raises expectations on conclusions about the microbarom model performance, the manuscript essentially evaluates the vespagram approach as a method to utilize infrasound for stratospheric diagnostics. Due to its capability to scan all directions simultaneously, the advantage of this novel approach is obvious. The manuscript is generally well written and organized, but the presentation and discussion of the results need to be enhanced. In particular, Section 3.1 confronts the reader with three extensive figures within a few sentences while remaining sparse with explaining and discussing details (e.g., on remarkable features, outliers) of these figures. For instance, Figure 4 is only once briefly referred to. My specific comments indicate several features that could or should be further discussed (e.g., Figs. 2 and 3).

Given the number of comments and questions, I request a major revision, rather than a minor. Once the authors have addressed these comments and questions, this study will be an important contribution and thus certainly worthy of publication in Annales Geophysicae.

Specific comments

1)  Consider revising the title of your manuscript a little (see general comment).
2)  l. 16: Why do microbaroms return to the ground after penetrating the middle atmosphere (hence their potential to probe the middle atmosphere dynamics)? Briefly explain the underlying physical process.
3)  l. 42: how did you determine the fixed apparent velocity of 350 m/s – from observations (using other processing techniques?) or propagation modeling (average?), or is this based on previous studies (references available? - obviously yes, but these are not cited before line 87/88). For the discussion of the results (e.g., line 205): using this fixed value, what is the corresponding standard deviation of observations at IS37 (e.g., using PMCC)? Based on this, can you roughly quantify the number of other arrivals (especially in the summer) that potentially cause discrepancies?
4)  l. 75 and Fig. 1a: you could add the ARCES array to the map as this is mentioned in the text as the initially planned site for IS37; however, I am wondering if the first part of the sentence ("was initially planned … in Karasjok") is worth to be mentioned at all. This fact is not relevant to your study but raises the question of why it was less favorable. Therefore I recommend shortening the paragraph accordingly.
5)  l. 125/126: it is not necessary to repeat all references, the choice of 350 m/s was justified before; I suggest removing the second part of the sentence (beginning with "which is within …").
6)  l. 136: Landès et al. (2014) studied the global patterns of microbaroms and only discuss the potential limitations due to the lack of coastal reflections while citing Hillers et al. (2012), among others. Therefore citing that study in the way it is done here is a bit misleading. My suggestion is to modify this and add another sentence, for example:

"Studies on microseisms (e.g., Hillers et al., 2012) have demonstrated the limitations of a model that does not account for coastal reflection. These limitations have been accordingly raised in the context of microbaroms (Landès et al., 2014)."

7) l. 153/154: which of the ECMWF models in particular? If not the ERA5 reanalysis, did you interpolate the temperature and wind fields in time?

8) l. 163: remove the parenthesis (private communication with …), M. De Carlo is co-author of this study. Instead, how would the results differ if you accounted for only 3000 km? (Is it essential to account for 5000 km for providing a more realistic spectrum at IS37?)

9) Section 3.1: Here you present a lot of information (3 figures within more than 24 panels!) within the first paragraph, without much explanation. You could help the reader by focusing on Figs. 2 and 3 first. Also, I suggest that you already define Eq. 3 in Section 2; then all panels can be understood at the first occurrence of a figure in Section 3.

10) According to Section 2.1 step 5, the vespa output should be power (Pa²), whereas in Figs. 2 and 3 the colorbar unit is Pascal (amplitude) again, correct? (also, place the units/labels to the right of the colorbars – amplitude in Pa).

11) Fig. 2b-d: in the summer, infrasound amplitudes at IS37 seem to be not relevant, whereas for the comparison (Fig. 2h) and through normalization (e-j) they certainly are (e.g., lower SI). Would a logarithmic color scale be useful in b)-d)? What is the impact of the detection threshold (noise level) of the station, especially for the summer season comparison – could this explain parts of the discrepancy between model and vespa in Fig. 2a?

12) Fig. 2j: One can recognize spots of maximum normalized power from south-easterly directions in the summer (not represented by the model). What could be their origin? There are probably not many potential sources in that direction (especially not for low frequencies).

13) Fig. 3d and particularly 3j: The vespagrams exhibit some horizontal lines (e.g., E and NW). Could these be artifacts of the vespa/beamforming processing?

14) l. 180 and Fig. 4: the median differences in direction of max. power are about 0-2 degree lower (by eye inspection) when using the smoothed model; the trend favoring the smoothed model is clearer recognized in the uncertainty ranges. However, if these uncertainties also correspond to the difference at the maximum power only, these are relatively large (not only at low frequency but also at the highest frequency band). How would you explain this?

15) Eq. 3: Please check if the equation is correctly noted. According to my understanding, the right-hand side is the definition of MSE(t). In this case, the equation should be modified to SI=1-MSE=1-(1/N)… or SI=1-MSE with MSE=(1/N)…

16) Eq. 3 / Figs. 2&3 / model output: The vespa analysis is done at a time step of 30min (1h time window), but the time step of the p2l data is 3h; do you interpolate the microbarom model output to 30min while smoothing or integrate the vespa over 3h? Do the time series in Figs. 2a and 3a (and b-g) differ in temporal resolution? What is the temporal resolution of the similarity index? Please briefly clarify in the manuscript.

17) l. 193-194: consider rephrasing this sentence towards SI instead of MSE; also, once SI has been defined in Section 2 (see comment 9), use SI for the axis labels in Figs. 2, 3, and 5, rather than 1-MSE.

18) l. 200: An SI of 0.5 corresponds to an MSE of 0.5, but the absolute difference between vespa and model must be even larger (and thus quite large!), due to the squared nature. In other words: For normalized distributions (within [0,1]), the MSE heavily weights small discrepancies instead of significant outliers, as opposed to when the absolute values exceed 1. Have you already contemplated using the mean absolute error instead?

19) Fig. 5: how are the data within a 3-day interval handled (mean/median, discrete)?
20) l. 239/240: "usually appears earlier" (3-24 hours) – this applies only to 2017 (and 2016), doesn't it?
21) l. 250: "resulting *in* model-vespagram discrepancies" – Can you quantify these discrepancies caused by ECMWF wind along the infrasound path?
22) l. 252/253: Do vespagrams perform better than other methods such as PMCC in the context of SSW events? I am aware that this is not your point here. Nevertheless, in other sections, you correctly highlight the advantage of the vespa approach (all directions simultaneously), whereas in Fig. 6 you compare the back-azimuths of the dominant signals only – which are likely similar to the output of PMCC.

Technical corrections

- De Carlo et al. (2020) reference: this is not unique, there are two entries in the list matching this citation! Add a/b letters.
- l. 8: revealed --> reveals
- l. 9 - add "events": sudden stratospheric warming [events].
- l. 16 - remove "back" (return or turn back are both appropriate, but return back looks like a tautology)
- l. 69 - Blanc et al. (2018) was referenced in the sentence before, could be saved here
- Fig. 1b - Sx/Sy = slowness components (I suggest you add this information to the caption, it is not defined in the text)
- l. 108: of the incoming signal
- l. 111 - remove "a" (or add a noun such as "approach" after "applied")
- l. 124: to the square root
- l. 134 - the WW3 reference is missing in the bibliography
- l. 136: […] as described by Ardhuin et al. (2011).
- l. 153: assess --> determine ("assess" is also used in the next sentence)
- l. 154: Forecasting --> Forecasts
- l. 167: resolution of array --> array resolution
- l. 174/175 - rephrasing suggestion: Figures 2a and 3a show the maximum amplitude per time step over one year, i.e. the dominant signals in the azimuthal spectra.
- l. 178: accompanied with --> accompanied by [the] (or: combined with the)
- l. 179 - "applying" is redundant
- Fig. 2/3 - j) should be g) in order to avoid confusion when reading the caption (e.g., e-j)
- Fig. 2 caption - I assume that panels 2-4, 6, and 7 are b-d, f, and j(g), correct?
- Fig. 2 caption: similarity score --> similarity index (Eq. 3)
- Fig. 2 caption: the colormap reference is also given in the acknowledgments of the manuscript; consider removing it from the caption to focus on the essentials.
- l. 188/189 - why do you use negative back-azimuths instead of 266° (265°), 239° (245°), and 26° (34°), respectively? Please also add the degree symbol (unit).
- Fig. 4 - please add a unit to the y label (°); the figure size could be smaller in the final version (width of one column)
- l. 202: in the Arctic
- l. 212: promising
- l. 213: the analysis
- Fig. 5 caption: Multi-year comparison between vespagrams and smoothed modelled microbarom soundscapes at the IS37 station.

- Fig. 5: could you include the legend of the last panel *inside* this panel? Consider using different colors for this panel.
- l. 235: […] until late March or early April, which corresponds […]
- Fig. 6 caption: days --> onset days
- l. 243: […] addressed by Diamantakis (2014) and Smets et al. (2016).
- l. 249: […] demonstrated by Evers and Siegmund (2009) and Smets and Evers (2014) that […]
- l. 270/271 - rephrase this sentence
- General technical remark: no space between number and % as well as °N, °E, …
- General grammatical remark: I think you should add articles to a number of nouns.

---

## Author Comment (AC1) · 16 Feb 2021

Dear Igor Chunchuzov, Thank you very much for your constructive review of the submission. We have made edits to the manuscript according to your comments and suggestions. Below, you can find our point-by-point reply to your report.

Specific comments: 1) The microbarom model used is based on microbarom generation model that predicts the spatial distribution of the acoustic sources over the ocean surface, and on the atmospheric model that allows one to calculate the microbarom propagation from the microbarom sources to the receivers. Each of these models has its own drawbacks, which introduce errors in the prediction of the parameters of

[Figure]

microbaroms at distances of thousands of kilometers from their sources. One of the drawbacks of the propagation model, which the authors themselves pointed out, is the approximation of a horizontally homogeneous atmosphere. The presence in the real atmosphere of the horizontal inhomogeneities in the wind velocity and temperature significantly affects the azimuth of arrival of the signal at the reception point and the prediction of the source back-azimuth. Indeed, the approximation of a horizontally homogeneous atmosphere has been made in the model. As you mention, we point this out in the manuscript, especially in the discussion section where we suggest different way to improve the results of simulations. To make the limitations of this approximation clearer to the reader, changes in Sect. 4 have been made.

2) Another disadvantage of the propagation model used is that the wind velocity and temperature profiles derived from the European Center for Medium range Weather Forecasting (ECMWF) do not have sufficient vertical resolution to account for the effect of small-scale atmospheric irregularities on microbarom scattering and, as a result, on amplitude attenuation with increasing distance from a source for different directions of propagation. This is a very good point. The ECMWF temperature and wind profiles indeed do not resolve small-scale irregularities in the atmosphere. And so far, resolving small-scale structures in atmospheric models, reanalysis and forecasting systems remains a topic for active research. On the contrary, the development and study of methods improving the resolution of atmospheric model's wind and temperature profiles using infrasonic observations are highly pertinent today (e.g. Chunchuzov et al., 2015; Amezcua et al., 2020; Rodriguez et al., 2020). However, the disadvantage you mention is relevant only if methods requiring atmospheric wind and temperature profiles as an input are used (such as the full waveform propagation modelling or 2D (3D) ray tracing). The semi-empirical attenuation law used in this study accounts for the Veff = V_50km/V_ground ratio, presenting atmospheric conditions above the station which are crucial for detecting the signal. Therefore, wind and temperature values at one specific level are used, and vertical resolution of the ECMWF is not significant. To make the limitations of the infrasound propagation modelling clearer, we have mentioned them

in Sect. 4.

3) When describing microbarom generation model the authors refer to the state-of-the-art microbarom radiation theory (De Carlo et al., 2020), which "...allows prediction of the location and intensity of the microbarom sources when applied to the Hasselmann integral." It would be important to note briefly in the paper of how are the frequency spectra of counter propagating waves derived in the wave model to calculate the Hasselmann integral, because the latter defines the distribution of the intensity of acoustic sources over ocean surface. Thank you for the valuable suggestion. More detailed description of the wave model used has been added into Section 2.2.

4) The parameters of microbaroms vs time were obtained for the fixed apparent velocity of 350 m/s, which corresponds to the arrivals of the signals from the stratospheric altitudes. Are there in the detected signal the microbarom reflections from the lower thermosphere with another apparent velocity? In our study calculations were performed for the fixed apparent velocity of 350 m/s, as you correctly note. To see if there are signals arriving from higher altitudes, for example from the lower thermosphere, the calculations need to be done for higher values of the apparent velocity (Lonzaga, 2015). These calculations are outside of the scope of the current research. However, Näsholm et al. (2020) demonstrated that mesospheric - lower thermospheric (MLT) arrivals originating from Iceland / Greenland hot-spot can be detected at IS37 in summer, but only if signal processing removes stratospheric arrivals from other directions such as Pacific / Barents Sea.

5) The amplitude obtained from the model in Fig.2a (red) in the time interval 200-201 DOY is two orders lower than the amplitude obtained by vespa processing. Could you explain the cause of such discrepancy? There could be various reasons explaining the discrepancy in Fig. 2a, e.g. an error in the wave model or in atmospheric winds causing an overestimation of the attenuation using the semi-empirical law. From Fig. 2f we can see that the modelled dominant direction is shifted a little bit towards the north when the discrepancy occurs, while there is no evident shift in Fig. 2j(g) for the vespagram.

Therefore, that could indeed be a wind issue, and we are looking at signals originating from different sources. The corresponding explanation has been added to Sect. 3.1.

6) Fig. 2f and Fig. 2j: Are the amplitudes (model and vespagram) in these Fig. s normalized by the maximum amplitude? Yes, Fig. 2 (e, f, j) and Fig. 3 (e, f, j) present amplitudes normalized by the maximum amplitude at each time step. We have now clarified this in Sect. 3.1 and in the caption of Fig. 2.

7) The last expression in the right side of (3) defines rather a mean squared error (MSE), than a similarity index (SI), since this expression becomes zero (not 1) in case of full match between model and infrasound vespagram. Thank you for the comment, the typo in the right side of (3) has been corrected. Now (3) is as follows: SI = 1 − MSE = 1 − (1/N) SUM (P_model − P_vespa)^2.

8) Line 185: "Going to higher frequencies, there is a pronounced change in the dominant direction of the source from the Atlantic in winter to the Barents Sea in summer (Fig. 3)." Do the higher frequencies react stronger on the change of wind direction in the stratosphere from eastward to westward than the lower ones? If yes, then why? In case of the low frequencies (0.1 − 0.2 Hz), there is a limited number of possible oceanic sources. To generate infrasound at such low frequencies, the source need to have a substantial size. In Fig. 2j(g) one can also see a change in the dominant source direction in summer. Signals coming from NE and SE are interpreted as those from the Pacific and the Indian oceans (see point 12 in the reply to R2 comments). However, this effect is more pronounced for the higher frequencies. The possible explanation could be the distance between IS37 and ocean sources. The North Atlantic microbarom source is located much closer to the station than the Pacific and the Indian oceans (~3000 km vs ~8000 km). Propagating over such a long distance, the attenuation might be crucial and lead to the signal to be below the noise threshold. This can also explain the reason why many data points have been ignored in the infrasound vespagram (Fig. 2j(g)) during summer (see point 11 in the reply to R2 comments).

Thank you for taking the time to review our submission, we believe that your advices have helped to clarify the manuscript.

Your sincerely, Ekaterina Vorobeva, on behalf of all authors

References Amezcua, J., Näsholm, S. P., Blixt, E. M., and Charlton-Perez, A. J.: Assimilation of atmospheric infrasound data to constrain tropospheric and stratospheric winds, Quarterly Journal of the Royal Meteorological Society, 2020.

Chunchuzov, I., Kulichkov, S., Perepelkin, V., Popov, O., Firstov, P., Assink, J., and Marchetti, E.: Study of the wind velocity-layered structure in the stratosphere, mesosphere, and lower thermosphere by using infrasound probing of the atmosphere, Journal of Geophysical Research: Atmospheres, 120, 8828–8840, 2015.

De Carlo, M., Ardhuin, F., and Le Pichon, A. (2020). Atmospheric infrasound generation by ocean waves in finite depth: unified theory and application to radiation patterns. Geophysical Journal International, 221, 569–585, https://doi.org/10.1093/gji/ggaa015

Lonzaga, J. B. (2015). A theoretical relation between the celerity and trace velocity of infrasonic phases. The Journal of the Acoustical Society of America, 138(3), EL242-EL247.

Näsholm, S. P., Vorobeva, E., Le Pichon, A., Orsolini, Y. J., Turquet, A. L., Hibbins, R. E., Espy, P. J., De Carlo, M., Assink, J. D., and Rodriguez, I. V. (2020). Semidiurnal tidal signatures in microbarom infrasound array measurements, in: EGU General Assembly Conference Abstracts.

Rodriguez, I. V., Näsholm, S. P., & Le Pichon, A. (2020). Atmospheric wind and temperature profiles inversion using infrasound: an ensemble model context. The Journal of the Acoustical Society of America, 148(5), 2923-2934.

Please also note the supplement to this comment:
https://angeo.copernicus.org/preprints/angeo-2020-78/angeo-2020-78-AC1-

supplement.pdf

---

## Author Comment (AC2) · 16 Feb 2021

Dear Referee 2, Thank you very much for your constructive review of the submission. We have made edits to the manuscript according to your comments and suggestions. Below, you can find our point-by-point reply to your report.

Specific comments: 1) Consider revising the title of your manuscript a little (see general comment). Thank you for the suggestion. The manuscript title has been changed to "Benchmarking microbarom radiation and propagation model against infrasound recordings: a vespagram-based approach".

[Figure]

2) l. 16: Why do microbaroms return to the ground after penetrating the middle atmosphere (hence their potential to probe the middle atmosphere dynamics)? Briefly explain the underlying physical process. The explanation of the infrasound waves refraction in the middle atmosphere has been added to Sect. 1.

3) l. 42: how did you determine the fixed apparent velocity of 350 m/s – from observations (using other processing techniques?) or propagation modeling (average?), or is this based on previous studies (references available? - obviously yes, but these are not cited before line 87/88). For the discussion of the results (e.g., line 205): using this fixed value, what is the corresponding standard deviation of observations at IS37 (e.g., using PMCC)? Based on this, can you roughly quantify the number of other arrivals (especially in the summer) that potentially cause discrepancies? This is a good point. An explanation of the choice of the fixed apparent velocity value has been added to Sect. 1. We find the comparison with the PMCC method to be beyond the scope of this article. Hence, all interpretations and explanations are based on discrepancies between the microbarom model outputs and vespagrams.

4) l. 75 and Fig. 1a: you could add the ARCES array to the map as this is mentioned in the text as the initially planned site for IS37; however, I am wondering if the first part of the sentence ("was initially planned . . . in Karasjok") is worth to be mentioned at all. This fact is not relevant to your study but raises the question of why it was less favorable. Therefore I recommend shortening the paragraph accordingly. Thank you for the suggestion. The sentence in l.75 has been changed following your comment.

5) l. 125/126: it is not necessary to repeat all references, the choice of 350 m/s was justified before; I suggest removing the second part of the sentence (beginning with "which is within . . ."). The sentence has been corrected according to the suggestion.

6) l. 136: Landès et al. (2014) studied the global patterns of microbaroms and only discuss the potential limitations due to the lack of coastal reflections while citing Hillers et al. (2012), among others. Therefore citing that study in the way it is done here
is a bit misleading. My suggestion is to modify this and add another sentence, for example:"Studies on microseisms (e.g., Hillers et al., 2012) have demonstrated the limitations of a model that does not account for coastal reflection. These limitations have been accordingly raised in the context of microbaroms (Landès et al., 2014)." Thank you for the comment. The paragraph has now been modified following your suggestion.

7) l. 153/154: which of the ECMWF models in particular? If not the ERA5 reanalysis, did you interpolate the temperature and wind fields in time? The ECMWF high Resolution (HRES) model has been used. The temporal resolution of this model is 6 hours which is twice WWIII time step. Therefore, to avoid possible discrepancy caused by interpolation in time, the assumption of the constant wind and temperature fields over 6 hours was made. Sect. 2.2 has been updated to clarify questions related to the ECMWF model used.

8) l. 163: remove the parenthesis (private communication with . . .), M. De Carlo is co-author of this study. Instead, how would the results differ if you accounted for only 3000 km? (Is it essential to account for 5000 km for providing a more realistic spectrum at IS37?) The choice of the maximum distance from the station depends on the location of the station and the main sources, as well as on how realistic spectrum is needed for a specific task. The recent study by De Carlo et al (2021) demonstrated a comparison of global microbarom patterns between the PMCC and the microbarom model by De Calo et al. (2020a) used in this study. The calculations have been performed using the maximum distance of 5000 km obtained from averaging over 45 IMS stations and providing more realistic spectra. The analysis reveal a good agreement between the PMCC and the model with the 5000 km cut off distance. Based on results of the aforementioned studies, we use the model configuration that provides the best estimate of microbarom spectra. Sect. 2.2 has been updated to clarify questions related to the choice of the maximum distance.

9) Section 3.1: Here you present a lot of information (3 figures within more than 24

panels!) within the first paragraph, without much explanation. You could help the reader by focusing on Figs. 2 and 3 first. Also, I suggest that you already define Eq. 3 in Section 2; then all panels can be understood at the first occurrence of a figure in Section 3. Thank you for the suggestion. Changes in Sect. 2 and Sect. 3.1 have been made according to your recommendation.

10) According to Section 2.1 step 5, the vespa output should be power ($Pa^2$), whereas in Figs. 2 and 3 the colorbar unit is Pascal (amplitude) again, correct? (also, place the units/labels to the right of the colorbars – amplitude in Pa). This is correct, the vespa output is power (Pa2). We have used Pascal unit in Figs. 2 and 3 hoping that this will help the reader to get an intuitive sense of the pressure amplitude. The corresponding explanation has now been added into Sect. 3.1 as well as in Fig. 2 caption. The units/labels in Figs. 2 and 3 has been moved to the right of the colorbars.

11) Fig. 2b-d: in the summer, infrasound amplitudes at IS37 seem to be not relevant, whereas for the comparison (Fig. 2h) and through normalization (e-j) they certainly are (e.g., lower SI). Would a logarithmic color scale be useful in b)-d)? What is the impact of the detection threshold (noise level) of the station, especially for the summer season comparison – could this explain parts of the discrepancy between model and vespa in Fig. 2a? In summer, infrasound amplitudes at IS37 are indeed lower than in winter. However, we believe they are still relevant. The normalization at every time step facilitates interpreting and comparing the directional spectra between data and model (Figs. 2 and 3 e – j (g now)). The main parameter influencing SI is the difference between the model's and vespagram's directional spectra. In winter, when atmospheric conditions are favorable for the stratospheric ducting from the west to the station, the assumption of a horizontally homogeneous atmosphere in the model doesn't affect the results as much as in summer, and the model and vespagrams are in a better agreement. However, in the summer or during SSW events, this assumption is not so valid and the effect of winds along the propagation path needs to be considered to a greater extent. This results in a large difference in directional spectra and, as a result, in lower

SI values. A corresponding explanation has been added to Sect. 3.1. See also point 14) for more details. We cannot directly account for the station detection level since we are calculating the power for different directions using a sliding time window – without applying trigger-based event detection approaches. However, after the vespa processing is done, we apply a quality check threshold based on the vespagram spectrum properties. At the time when vespa processing predicts a directional spectrum with the power (almost) equal in all directions, data are ignored. This is especially pronounced for the 0.1 – 0.2 Hz band during summer (see Fig. 2 g). Changes in Fig. 2 has now been made in order to highlight the lack of data in the summertime.

12) Fig. 2j: One can recognize spots of maximum normalized power from southeasterly directions in the summer (not represented by the model). What could be their origin? There are probably not many potential sources in that direction (especially not for low frequencies). You are right, there are not so many potential sources in SE direction that would provide so low frequency microbaroms. This could be microbaroms generated in the Indian ocean. The microbarom model's map for June 2016 supports this hypothesis (see the supplement attached). The stratospheric summertime westward wind could guide the infrasound waves towards the IS37 station. The distance between the station under consideration and the Indian ocean is much larger than 5000 km (around 7000 – 8000 km) which is the model's cut off limit. Therefore, these arrivals are not presented by the model. The corresponding explanation has been added to the discussion of Fig. 2.

13) Fig. 3d and particularly 3j: The vespagrams exhibit some horizontal lines (e.g., E and NW). Could these be artifacts of the vespa/beamforming processing? Indeed, sidelobes in the steered response can appear as an inherent effect of array geometry. Still, it is maybe not so straightforward to find the source of those lines. Additional power peaks that arise in the vespa processing represent side lobes appearing when extracting power values along the fixed apparent velocity circle (Fig. 1b). As can be seen from Fig. 1b, for lower frequencies we have less side lobes. Since Fig. 3
considers the 0.5 – 0.6 Hz band, the number of side lobes is higher, but their amplitude is several dB lower than the main lobe. More importantly, for any side lobes related effect, the position of the "lines" would change over time, staying approximately at the same angular distance from the dominant signal direction. Therefore, we lean towards not believing this is a result of the vespa processing and assume that those lines could present some stable local background sources of infrasound with frequencies within the microbarom range.

14) l. 180 and Fig. 4: the median differences in direction of max. power are about 0-2 degree lower (by eye inspection) when using the smoothed model; the trend favoring the smoothed model is clearer recognized in the uncertainty ranges. However, if these uncertainties also correspond to the difference at the maximum power only, these are relatively large (not only at low frequency but also at the highest frequency band). How would you explain this? Thank you for this question. Both medians and uncertainty ranges in Fig. 4 are estimated based on the back-azimuth difference at the maximum power only. Thanks to your request, we have checked the calculation procedure and found an error in the calculation for the lowest frequency band. The calculation results for the remaining frequency bands remain unchanged. An updated version of Fig. 4 can be found in the manuscript. Uncertainty values falling between 25 and 75 percentiles are an objective assessment of the discrepancy between the model and vespagrams. These values originate from the wintertime when atmospheric conditions are favorable for stratospheric ducting from the West. In summer, atmospheric conditions are not so stable and there are several factors that can cause discrepancies as we mention in l. 200 – 210. The vespagram-based approach, in turn, is very sensitive to atmospheric changes opposite to the model which uses only atmospheric conditions at the station to access the possibility of a wave front arrival. Therefore, summer arrivals predicted by the model look more stable than those predicted by vespagrams (see Fig. 3). The difference between the direction of max in the summertime can reach up to tens of degrees, for example when the model predicts arrivals from the Barents Sea and the vespagram predicts arrivals from the North Atlantic (Fig. 3 around day 210 in
2016). This also causes a fall of the similarity index. A corresponding explanation has been added to Sect. 3.1.

15) Eq. 3: Please check if the equation is correctly noted. According to my understanding, the right-hand side is the definition of MSE(t). In this case, the equation should be modified to SI=1-MSE=1-(1/N)... or SI=1-MSE with MSE=(1/N)... Thank you for the comment, the typo in the right side of (3) has been corrected. Now (3) is as follows: SI = 1 − MSE = 1 − (1/N) ïĄŞ (P_model − P_vespa)^2.

16) Eq. 3 / Figs. 2&3 / model output: The vespa analysis is done at a time step of 30min (1h time window), but the time step of the p2l data is 3h; do you interpolate the microbarom model output to 30min while smoothing or integrate the vespa over 3h? Do the time series in Figs. 2a and 3a (and b-g) differ in temporal resolution? What is the temporal resolution of the similarity index? Please briefly clarify in the manuscript. Thank you for this question. In this study, in order to avoid the model output's interpolation in time, the vespa processing output has been sub-sampled to match the three hourly microbarom model grid. Further, all results are presented with the time resolution of 3 h. The corresponding description has been added to Sect. 2.3.

17) l. 193-194: consider rephrasing this sentence towards SI instead of MSE; also, once SI has been defined in Section 2 (see comment 9), use SI for the axis labels in Figs. 2, 3, and 5, rather than 1-MSE. Corresponding corrections have been made in Figs. 2, 3, 5 as well as in Sect. 2.3.

18) l. 200: An SI of 0.5 corresponds to an MSE of 0.5, but the absolute difference between vespa and model must be even larger (and thus quite large!), due to the squared nature. In other words: For normalized distributions (within [0,1]), the MSE heavily weights small discrepancies instead of significant outliers, as opposed to when the absolute values exceed 1. Have you already contemplated using the mean absolute error instead? The calculation of SI based on normalized distributions is justified by the significant effect of smoothing procedure on modelled amplitudes. Comparison

of the unsmoothed model with the vespa calculation results is not used, because the model does not account for the frequency-dependent resolution of the infrasonic array. The mean square error calculation is a widely used approach that allows a comparison between two statistical models. Therefore, MSE can represent the difference between the actual observations and the observation values predicted by the model. Following your advice, we have now explored using the mean absolute error (MAE) instead. The main conclusion from that experiment is that using the MAE doesn't significantly change the results or conclusions based on them. For this reason, the SI calculation procedure has not been changed in our manuscript.

19) Fig. 5: how are the data within a 3-day interval handled (mean/median, discrete)? The data in Fig. 5 are presented as a discrete set with 3-day step, namely, day 0 00 hours, day 3 00 hours etc. The median is presented in the last panel only. The corresponding explanation has been added to the manuscript.

20) l. 239/240: "usually appears earlier" (3-24 hours) – this applies only to 2017 (and 2016), doesn't it? This applies to all years under consideration depending on the frequency band used. Fig. 6 presents the results for 0.3 – 0.4 Hz band where, as you correctly mention, this applies to 2016 and 2017. The corresponding explanation has been added to Sect. 3.2.

21) l. 250: "resulting *in* model-vespagram discrepancies" – Can you quantify these discrepancies caused by ECMWF wind along the infrasound path? These discrepancies have been already quantified in l. 238. L. 250 has been changed to "resulting in the above-mentioned model-vespagram discrepancies".

22) l. 252/253: Do vespagrams perform better than other methods such as PMCC in the context of SSW events? I am aware that this is not your point here. Nevertheless, in other sections, you correctly highlight the advantage of the vespa approach (all directions simultaneously), whereas in Fig. 6 you compare the back-azimuths of the dominant signals only – which are likely similar to the output of PMCC. As mentioned

in the initial part of Sect. 3.2, studying the behavior of SSW events is not the main objective of the study. The main point of this section is rather to examine the ability of the vespagrams to detect extreme atmospheric events and see if there are significant discrepancies with the model. We considered changes in the back-azimuths of the dominant signals only in Fig. 6 because this is one of the infrasound signatures of SSW events. Moreover, such approach is one of the few ways to present vespagram and model output in the same plot. It is impossible to present two colorbar plots in one. However, trying to follow your advice, updates have been made in Fig. 6. The figure now presents the microbarom azimuthal distribution at IS37 estimated by vespa and normalized per time step, as well as the back-azimuths of the dominant signals predicted by the model (red dots). Comparison of the vespagrams and the PMCC is not within the scope of the current study but could serve as an idea for future studies. The model by De Carlo et al. (2020a) has already been compared with the output of PMCC for multiple stations including IS37 (De Carlo et al., 2021).

Technical corrections: - De Carlo et al. (2020) reference: this is not unique, there are two entries in the list matching this citation! Add a/b letters. Thank you for spotting this. We have made the associated corrections.

- l. 8: revealed –> reveals Corrected.

- l. 9 - add "events": sudden stratospheric warming [events]. Corrected.

- l. 16 - remove "back" (return or turn back are both appropriate, but return back looks like a tautology) Corrected.

- l. 69 - Blanc et al. (2018) was referenced in the sentence before, could be saved here The reference has been removed from l. 69.

- Fig. 1b - Sx/Sy = slowness components (I suggest you add this information to the caption, it is not defined in the text) A definition of Sx/Sy has now been added to the caption of Fig. 1.

- l. 108: of the incoming signal Corrected.

- l. 111 - remove "a" (or add a noun such as "approach" after "applied") The article "a" has been removed.

- l. 124: to the square root Changed according to the suggestion.

- l. 134 - the WW3 reference is missing in the bibliography The reference has been added to the bibliography.

- l. 136: [...] as described by Ardhuin et al. (2011). Corrected.

- l. 153: assess –> determine ("assess" is also used in the next sentence) Changed.

- l. 154: Forecasting –> Forecasts Corrected.

- l. 167: resolution of array –> array resolution Corrected.

- l. 174/175 - rephrasing suggestion: Figures 2a and 3a show the maximum amplitude per time step over one year, i.e. the dominant signals in the azimuthal spectra. This sentence has been rephrasing according to your suggestion.

- l. 178: accompanied with –> accompanied by [the] (or: combined with the) The phrase "accompanied with" has been replaced with "accompanied by the" following the suggestion.

- l. 179 - "applying" is redundant The word "applying" has been removed from the sentence.

- Fig. 2/3 - j) should be g) in order to avoid confusion when reading the caption (e.g., e-j) Thank you. The index j) in Figs. 2 - 3 have been changed to g).

- Fig. 2 caption - I assume that panels 2-4, 6, and 7 are b-d, f, and j(g), correct? This is correct. The numbers in the caption have now been replaced with the letters to avoid confusion.

- Fig. 2 caption: similarity score –> similarity index (Eq. 3) This caption has now been

corrected.

- Fig. 2 caption: the colormap reference is also given in the acknowledgments of the manuscript; consider removing it from the caption to focus on the essentials. This reference was initially only mentioned in the acknowledgements. However, during the preprocessing of the manuscript by the journal, the editorial support team kindly asked us to add the image credit to the corresponding figure caption(s). Hence, although we would prefer to follow your advice, we opt to stay with what was requested by the editorial. Still, in order to avoid repetition, we have now removed this reference from the acknowledgments.

- l. 188/189 - why do you use negative back-azimuths instead of 266° (265°), 239° (245°), and 26° (34°), respectively? Please also add the degree symbol (unit). The negative values of the back-azimuth have been used in order to keep 0° (or the North) in the middle of (-180°, 180°) interval. However, following your advice, we have changed negative values to positive and have added the degree symbol.

- Fig. 4 - please add a unit to the y label (°); the figure size could be smaller in the final version (width of one column) Thank you for the recommendation. The unit (°) has been added to the y label in Fig. 4. The figure size will be changed by the editorial office later.

- l. 202: in the Arctic Corrected.

- l. 212: promising Corrected.

- l. 213: the analysis Corrected.

- Fig. 5 caption: Multi-year comparison between vespagrams and smoothed modelled microbarom soundscapes at the IS37 station. The caption has been changed according to the suggestion.

- Fig. 5: could you include the legend of the last panel *inside* this panel? Consider using different colors for this panel. Fig. 5 has been updated following your suggestion.

- l. 235: [. . .] until late March or early April, which corresponds [. . .] Corrected according to the suggestion.

- Fig. 6 caption: days –> onset days Changed.

- l. 243: [. . .] addressed by Diamantakis (2014) and Smets et al. (2016). Corrected.

- l. 249: [. . .] demonstrated by Evers and Siegmund (2009) and Smets and Evers (2014) that [. . .] Corrected.

- l. 270/271 - rephrase this sentence The sentence has been rephrased.

- General technical remark: no space between number and % as well as °N, °E, . . . We consulted the journal guidelines at https://www.annales-geophysicae.net/submission.html#manuscriptcomposition ), and found that manuscripts shall include a space between number and % as well as between ° and N. Looking at the final typeset version of other ANGEO papers, it looks like these have a reduced-width blank between number and symbol – so this will hopefully come out visually pleasing also in our final product.

- General grammatical remark: I think you should add articles to a number of nouns. Thanks for this advice. The grammar has been double-checked.

Thank you for taking the time to review our submission, we believe that your advices have helped to clarify the manuscript.

Your sincerely, Ekaterina Vorobeva, on behalf of all authors

References

De Carlo, M., Ardhuin, F., and Le Pichon, A.: Atmospheric infrasound generation by ocean waves in finite depth: unified theory and application to radiation patterns, Geophysical Journal International, 221, 569–585, https://doi.org/10.1093/gji/ggaa015, 2020a

De Carlo, M., Hupe, P., Le Pichon, A., Ceranna, L., and Ardhuin, F.: Global Microbarom Patterns: a First Confirmation of the Theory for Source and Propagation, Geophysical Research Letters, 48, e2020GL090 163, https://doi.org/https://doi.org/10.1029/2020GL090163, 2021

Please also note the supplement to this comment:
https://angeo.copernicus.org/preprints/angeo-2020-78/angeo-2020-78-AC2-supplement.pdf

---

## Referee Report (RR1)

Referee comment: "Benchmarking microbarom radiation and propagation model against infrasound recordings: a vespagram-based approach" (Vorobeva et al.)

General comments:

I thank the authors for responding to all comments adequately. The authors have done a great job in revising their manuscript. They enhanced the figures (particularly 2, 3, and 6) and rearranged parts of the manuscript, so the presentation of the methods and results is improved. Also, the concluding discussion is more comprehensive now. These improvements have strengthened my opinion that this manuscript, which presents a novel method for infrasound data analysis, will be a significant contribution to the infrasound community.

Therefore the manuscript can be accepted for publication in Annales Geophysicae subject to minor revisions. Please find specific comments and technical suggestions in the attachment.

Specific comments (line number refer to the cleaned version)

1) L. 166: "a more accurate simulation" – compared to what? (semi-empirical attenuation law, I guess, but this is mentioned in the next sentence)
2) L. 173: I suggest to add "… (HRES) model [analysis]…" in order clarify that neither a reanalysis nor a HRES forecast is used here; also in L. 175: "… ECMWF HRES [analysis] is 6 h"
3) L. 176, rather just a remark: a discrepancy can also be caused by the assumption of constant wind/temperature over 6 hours.
4) L. 202: "not so stable" – in which sense? direction?
5) Figs. 2, 3: b)-d) add the unit Pascal to the colorbar, f)-g) same colorbar as in e)?
6) L. 240/241: How much greater? (distances of ~8000 km?)
7) L. 293: only "from reanalysis data"? (according to my understanding, you do not use reanalysis data in your study), maybe you can write "from (re-)analysis data"
8) Fig. 6: This figure is nice, better than before! I suggest adding one colorbar (for the vespa) and changing the backazimuth to 0-360° (as you did in the text).

Technical corrections (line number refer to the cleaned version)

- L. 127: wavefront -> wave front (as elsewhere in the manuscript)
- L. 148: You start three subsequent sentences with "This …"  The second sentence could be rephrased, e.g.: "Here, we use …" (or passive mode)
- L. 168/169: "this law" (2x)
- L. 177: "… is [an] acoustic spectra attenuated …"
- L. 182: "on how realistic [a] spectrum is needed for…" (?)
- L. 198: a difference -> differences
- L. 221-222: "accompanied by the semi-empirical wave attenuation law" – here, I recommend using "combined with" or "complemented by" as you use "accompanied by" a few sentences earlier in a completely different context (where it fits better in my opinion)

- L. 223: "… between day[s] 200 and 210, [when] the modelled amplitude [is] much lower …"
- L. 234/235, rephrase: "However, the maximum power is sometimes also observed from north-easterly and south-easterly directions in summer".
- L. 247: "… and has values [close to] one, …"
- Fig. 5 caption, suggestion for better readability: "day 1 – 00 UTC, day 4 – 00 UTC, etc."
- Fig. 5: "depending on [the] year"
- L. 301: "[does] not always characterize"
- L. 303: difference[s]
- L. 323/333: disturbances instead of irregularities?

---

## Author Response (AR2)

Response to the comments posted on 5 May 2021 by anonymous Referee 2

**Benchmarking microbarom radiation and propagation model against infrasound recordings: a vespagram-based approach**

By Ekaterina Vorobeva, Marine De Carlo, Alexis Le Pichon, Patrick Joseph Espy, and Sven Peter Näsholm

Manuscript ID angeo-2020-78

Dear Topical Editor Dr Stober,

Many thanks for your time and for overseeing the resubmission processing and review. Please see below for our edits made in response to the reviewer report. In addition to these modifications, we have made a few minor typographic fixes and wording corrections in Sect. 2.2. which are also visible in our track changes PDF [ lines ~145; line 165; line 168].

Dear Referee 2,

Thank you very much for your constructive review of the revised manuscript. We have made edits according to your comments and suggestions. Below, you can find our point-by-point reply to your report.

**Specific comments**

1) *L. 166: "a more accurate simulation" – compared to what? (semi-empirical attenuation law, I guess, but this is mentioned in the next sentence)*
The phrase "a more accurate simulation" has been changed to "an accurate simulation" to avoid comparison in the sentence.

2) *L. 173: I suggest to add "… (HRES) model [analysis]…" in order clarify that neither a reanalysis nor a HRES forecast is used here; also in L. 175: "… ECMWF HRES [analysis] is 6 h"*
We clarified the atmospheric model specification used in the study.

3) *L. 176, rather just a remark: a discrepancy can also be caused by the assumption of constant wind/temperature over 6 hours.*
Thank you for the remark.

4) *L. 202: "not so stable" – in which sense? direction?*
We clarified the sentence in line 202.

5) *Figs. 2, 3: b)-d) add the unit Pascal to the colorbar, f)-g) same colorbar as in e)?*
The unit label has been added to the relevant colorbars of Figs. 3 and 4. The caption of Fig. 3 has been changed to clarify that panels b) – d) have a common colorbar, the same applies to panels e) – g).

6) *L. 240/241: How much greater? (distances of ~8000 km?)*
We clarified the sentence in lines 240/241.

7) *L. 293: only "from reanalysis data"? (according to my understanding, you do not use reanalysis data in your study), maybe you can write "from (re-)analysis data"*
Corrected following your suggestion.

8) *Fig. 6: This figure is nice, better than before! I suggest adding one colorbar (for the vespa) and changing the backazimuth to 0-360° (as you did in the text).*
Fig. 6 has been changed according to your suggestions.

**Technical corrections**

*- L. 127: wavefront -> wave front (as elsewhere in the manuscript)*
Corrected.

*- L. 148: You start three subsequent sentences with "This …" The second sentence could be rephrased, e.g.: "Here, we use …" (or passive mode)*
The paragraph has been rephrased.

*- L. 168/169: "this law" (2x)*
Corrected.

*- L. 177: "… is [an] acoustic spectra attenuated …"*
Corrected.

*- L. 182: "on how realistic [a] spectrum is needed for…" (?)*
The quality of spectrum simulated depends on number of sources taken into account. Shorter cut-off distance – fewer sources considered – less realistic spectra are obtained. To compare the model with the vespa processing we need to obtain as accurate spectra as possible. Previous studies demonstrated that 5000 km limit is the best candidate for that.

*- L. 198: a difference -> differences*
Corrected.

*- L. 221-222: "accompanied by the semi-empirical wave attenuation law" – here, I recommend using "combined with" or "complemented by" as you use "accompanied by" a few sentences earlier in a completely different context (where it fits better in my opinion)*
The part " accompanied by" has been changed to "combined with" following your suggestion.

*- L. 223: "… between day[s] 200 and 210, [when] the modelled amplitude [is] much lower …"*
Corrected.

*- L. 234/235, rephrase: "However, the maximum power is sometimes also observed from north-easterly and south-easterly directions in summer".*
The sentence has been rephrased.

*- L. 247: "… and has values [close to] one, …"*
Corrected.

*- Fig. 5 caption, suggestion for better readability: "day 1 – 00 UTC, day 4 – 00 UTC, etc."*
Corrected.

*- Fig. 5: "depending on [the] year"*
Corrected.

*- L. 301: "[does] not always characterize"*
Corrected.

*- L. 303: difference[s]*
Corrected.

*- L. 323/333: disturbances instead of irregularities?*
The word "irregularities" has been replaced with "disturbances".

Thank you again for taking the time to review our submission, we believe that your advice has helped to clarify the manuscript.

Your sincerely,

Ekaterina Vorobeva, on behalf of all authors